# Rule-based omics mining reveals antimicrobial macrocyclic peptides against drug-resistant clinical isolates

Zhuo Cheng[1,2], Bei-Bei He [1], Kangfan Lei[3], Ying Gao [1], Yuqi Shi[1], Zheng Zhong [1], Hongyan Liu[1], Runze Liu[1], Haili Zhang[2], Song Wu [3], Wenxuan Zhang [3] ✉, Xiaoyu Tang [2] ✉ & Yong-Xin Li [1] ✉

Antimicrobial resistance remains a significant global threat, driving up mortality rates worldwide. Ribosomally synthesized and post-translationally modified peptides have emerged as a promising source of novel peptide antibiotics due to their diverse chemical structures. Here, we report the discovery of new aminovinyl-(methyl)cysteine (Avi(Me)Cys)-containing peptide antibiotics through a synergistic approach combining biosynthetic rule-based omics mining and heterologous expression. We first bioinformatically identify 1172 RiPP biosynthetic gene clusters (BGCs) responsible for Avi(Me)Cys-containing peptides formation from a vast pool of over 50,000 bacterial genomes. Subsequently, we successfully establish the connection between three identified BGCs and the biosynthesis of five peptide antibiotics via biosynthetic rule-guided metabolic analysis. Notably, we discover a class V lanthipeptide, massatide A, which displays excellent activity against gram-positive pathogens, including drug-resistant clinical isolates like linezolid-resistant *S. aureus* and methicillin-resistant *S. aureus*, with a minimum inhibitory concentration of 0.25 μg/mL. The remarkable performance of massatide A in an animal infection model, coupled with a relatively low risk of resistance and favorable safety profile, positions it as a promising candidate for antibiotic development. Our study highlights the potential of Avi(Me)Cys-containing peptides in expanding the arsenal of antibiotics against multi-drug-resistant bacteria, offering promising drug leads in the ongoing battle against infectious diseases.

The rise of multi-drug-resistant bacteria, often referred to as "superbugs", has become an alarming global health crisis, resulting in infections that are increasingly difficult to treat[1]. The urgent need for new antibiotics to combat these superbugs is indisputable. In addition to small-molecule antibiotics, macrocyclic peptides serve as a complementary alternative for lead discovery. Macrocyclic peptides, which include both nonribosomal peptides (NRPs) and ribosomally synthesized post-translationally modified peptides (RiPPs), have emerged as a promising source for antibiotic discovery[2]. Their favorable properties, achieved through their unique macrocyclic structure, enable them to resist proteolysis and maintain an active conformation for effective target binding[3]. Noteworthy RiPP antibiotics, like nisin[4],

[1]Department of Chemistry and The Swire Institute of Marine Science, The University of Hong Kong, Pokfulam Road, Hong Kong, China. [2]Institute of Chemical Biology, Shenzhen Bay Laboratory, Shenzhen 515832, China. [3]State Key Laboratory of Bioactive Substance and Function of Natural Medicines, Institute of Materia Medica, Chinese Academy of Medical Sciences and Peking Union Medical College, Beijing 100050, China. ✉e-mail: wxzhang@imm.ac.cn; xtang@szbl.ac.cn; yxpli@hku.hk

ruminococcin[5], thiostrepton[6], nosiheptide[7], darobactin[8] and dynobactin[9], as well as NRPs, such as vancomycin[10], teixobactin[11], brevicidine[12], rimomycin[13], misaugamycin[13], corbomycin[14], and clovibactin[15], underscore the considerable potential of macrocyclic peptides in fighting against bacterial infections.

The traditional chemical and/or biological screening method in uncovering new antibiotics from field-collected resources or laboratory-cultured organisms has proven to be both time-consuming and inadequate[1,2]. Recent advances in sequencing techniques have generated a large number of microbial genomes in the publicly available database, providing new opportunities for the exploration of novel peptide antibiotics through genome mining[16]. Among the vast array of natural products, RiPPs stand out as a particularly expansive and diverse family, offering a remarkable diversity of structures and bioactive potential for antibiotic discovery[17,18]. As a result, substantial efforts have been devoted to genome mining-guided discovery of new RiPP antibiotics[19,20]. In recent studies, our research has focused on genomics-guided discovery of RiPP antibiotics, leading to the discovery of the first lanthipeptide from archaea with antagonistic activity and narrow-spectrum class II bacteriocins from the human microbiome[21,22]. By applying sequence- and 3D-structure-based genome mining strategies, we have successfully revealed previously hidden peptidases[23] and untapped post-translational modification (PTM) enzymes[24,25] involved in RiPP antibiotic biosynthesis. Nevertheless, the challenge remains in prioritizing biosynthetic gene clusters (BGCs) of interest and linking them to metabolites and their antibiotic potential prior to time-consuming isolation and identification[16,26–29].

Aminovinyl-(methyl-)cysteine-containing peptides (ACyPs) represent an attractive class of compounds with distinctive cyclic peptide rings and notable antibacterial properties[30]. The presence of the Avi(Me)Cys unit within ACyPs imparts structural rigidity, which restricts conformational flexibility, thereby protecting these peptides from proteolysis[30]. This highly modified structure also confers drug-like properties to the compound, including heat stability and high target specificity[30]. Microbisporicin, an Avi(Me)Cys-containing lanthipeptide, interacts electrostatically with the negatively charged lipid II pyrophosphate bridge, making them effective against vancomycin-resistant bacteria[31,32]. Similarly, lexapeptide exhibits potent activity against methicillin-resistant *Staphylococcus aureus* (MRSA) and methicillin-resistant *Staphylococcus epidermidis* (MRSE)[33]. To date, the Avi(Me)Cys moiety has been found in five RiPP families, including lanthipeptides, lipolanthins, lanthidins, thioamitides, and linaridins[30,34–39] (Fig. 1a and S1). The genetically encoded nature and diverse biosynthesis pathways of ACyPs allow for the efficient expansion of their chemical space. Despite the structural diversity of ACyPs resulting from various modifications, such as the methylation in linaridin[34], fatty acid chain in lipolanthin[37,38] and thioamide bond in thioamitide[35,40], the biosynthetic machinery towards the Avi(Me)Cys unit is remarkably conserved (Fig. 1a, b and S1). A characteristic flavoprotein catalyzes the oxidative decarboxylation of precursor C-terminal cysteine, yielding enethiol. Subsequently, thioether cyclization with dehydroalanine (Dha) or dehydrobutyrine (Dhb) leads to the generation of AviCys or AviMeCys[30,41–44] (Fig. 1b). Considering their unique structural features, promising bioactivities, and the increasing availability of microbial genomes, our motivation lies in conducting a systematic bioinformatic investigation of underexploited ACyPs to discover novel antibiotics.

Here, we conduct a large-scale analysis of publicly available actinobacteria and firmicutes genomes building upon the conserved biosynthetic logic of ACyPs, using the rule-based genome mining pipeline SPECO[24] (Fig. 1b, Supplementary data 1). Through sequence similarity network (SSN) and sequence logo analysis, we uncover 1172 BGCs with the potential to produce ACyPs, which could be categorized into 67 ACyP families. By implementing a mass mapping pipeline, we efficiently link five ACyPs to their corresponding BGCs

following a biosynthetic rule-based metabolomic analysis. This allows us to swiftly identify the relevant BGCs before proceeding with the isolation process. This is further substantiated through the successful heterologous expression of BGCs in a *Streptomyces* host. Furthermore, our bioactivity assay demonstrates that massatide A exhibits exceptional potency against various gram-positive bacteria, as evidenced by their strong inhibitory activity both in vitro and in vivo. These findings present promising avenues for the identification and characterization of ACyPs, which could serve as a valuable source of new antibiotics to combat the increasing prevalence of multi-drug-resistant bacteria.

## Results

### SPECO-based genome mining expands Avi(Me)Cys-containing RiPP families

Ribosomally synthesized post-translationally modified peptides are biosynthesized using a conserved mechanism in which post-translational modification (PTM) enzymes modify a ribosomal precursor peptide, followed by protease-mediated trimming to release the mature product[18]. Similarly, ACyP biosynthesis relies on a flavoprotein (Pfam number: PF02441) that catalyzes the oxidative decarboxylation of the C-terminal cysteine residue of the precursor. This process leads to the formation of an enethiol group, which can react with unsaturated amino acids, facilitated by oxidative thioaldehyde formation of the cysteine residue and parallel decarboxylation through tautomerization[30,41–44]. The requirement of the flavoprotein for thioenol formation and the indispensability of the C-terminal cysteine residue in precursor peptides provide a unified rule for genome mining of ACyP-encoding BGCs (Fig. 1b). We noticed that ACyPs were mainly found from firmicutes and actinobacteria, the genomes from these two phyla were then selected and analyzed.

Following this rule, we analyzed 21,911 genomes of actinobacteria and 30,666 genomes of firmicutes for the ACyP BGC identification using the SPECO pipeline[24] (Fig. 1b). This analysis uncovered 1172 unique putative precursor-flavoprotein pairs (Supplementary data 1). Our findings indicate that most putative precursors were not clustered with known precursors (Fig. 2a) as shown in the precursor sequence similarity network (SSN), highlighting the unexplored chemical space of ACyPs. As expected, known precursors such as epidermin, thioviridamide, lexapeptide, and cypemycin precursors were included in the top rank (Fig. 2a and S2). We further analyzed the putative BGCs through sequence logos (Supplementary data 2) and phylogenetic analysis (Supplementary data 3) to prioritize candidates with unknown precursors or tailoring enzymes. Among the identified BGC families, one family caught our attention due to the presence of two oxidoreductases, which is uncommon in ACyP BGCs (Fig. 2b). Further analysis of the genomic context revealed that multiple modifying enzymes were well conserved in this family. We named this family the *mat* family and hypothesized that the additional oxidoreductase might introduce alternative modifications to the precursor peptide. We were also drawn to the largest cluster in the SSN, which contains pristinin A3[45], and have designated this cluster as *sis* family. Similar to the *mat* family, the sequence logos of the *sis* family exhibited a characteristic C-terminal Avi(Me)Cys ring formation motif (T-X1-X2-X3-X4-C, T is threonine, C is cysteine, X represents a variable residue), and an additional conserved cysteine at the N-terminus (Fig. 2c). However, analysis of the precursor sequence logo for the *sis* family revealed an approximately 50% occurrence of Cys residues in the middle of the core region, which could be further categorized into two subfamilies with either two or three Cys residues (Fig. 2c). We propose that a single maturation system within the *sis* family could generate ACyPs with diverse thioether rings, owing to the propensity of Cys residues to react with Dha or Dhb. Overall, our analysis prioritized potential BGC candidates for exploring the chemical diversity of Avi(Me)Cys-containing RiPPs.

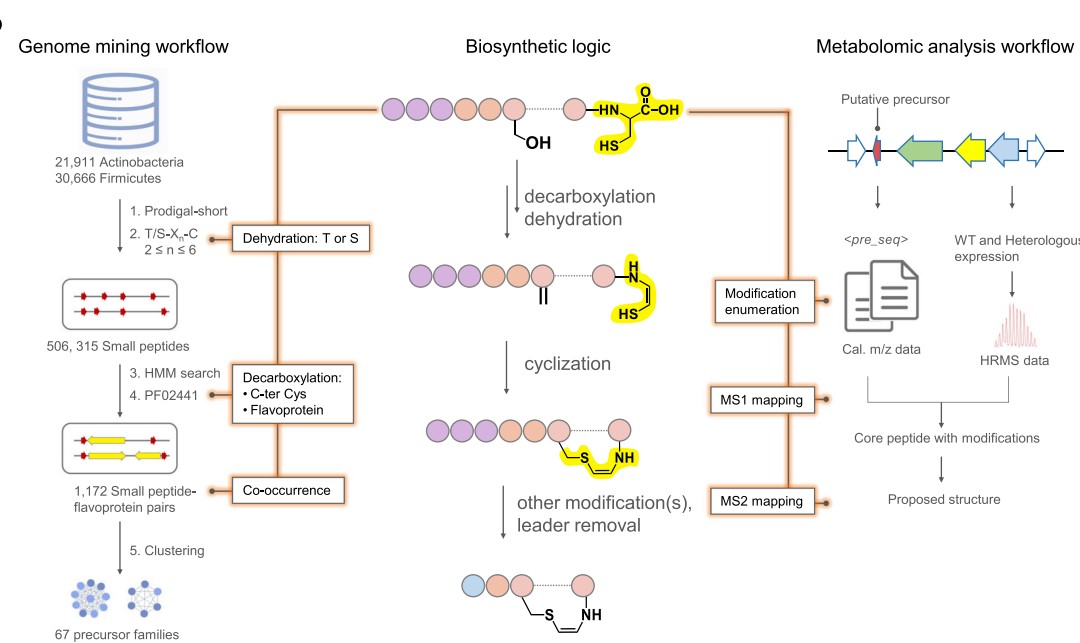

**Fig. 1 | Representatives, biosynthesis, and genome mining of aminovinyl-cysteine-containing peptides. a** The structure of Avi(Me)Cys motif and representative examples of Avi(Me)Cys-containing RiPPs, with the Avi(Me)Cys motifs highlighted in yellow. **b** Overview of the rule-based omics mining workflow: The central panel illustrates the common biosynthetic process for Avi(Me)Cys unit formation, which informs the genome mining of Avi(Me)Cys-containing RiPP BGCs using SPECO (left panel) and guides the metabolomics analysis (right panel) for targeted peptide identification.

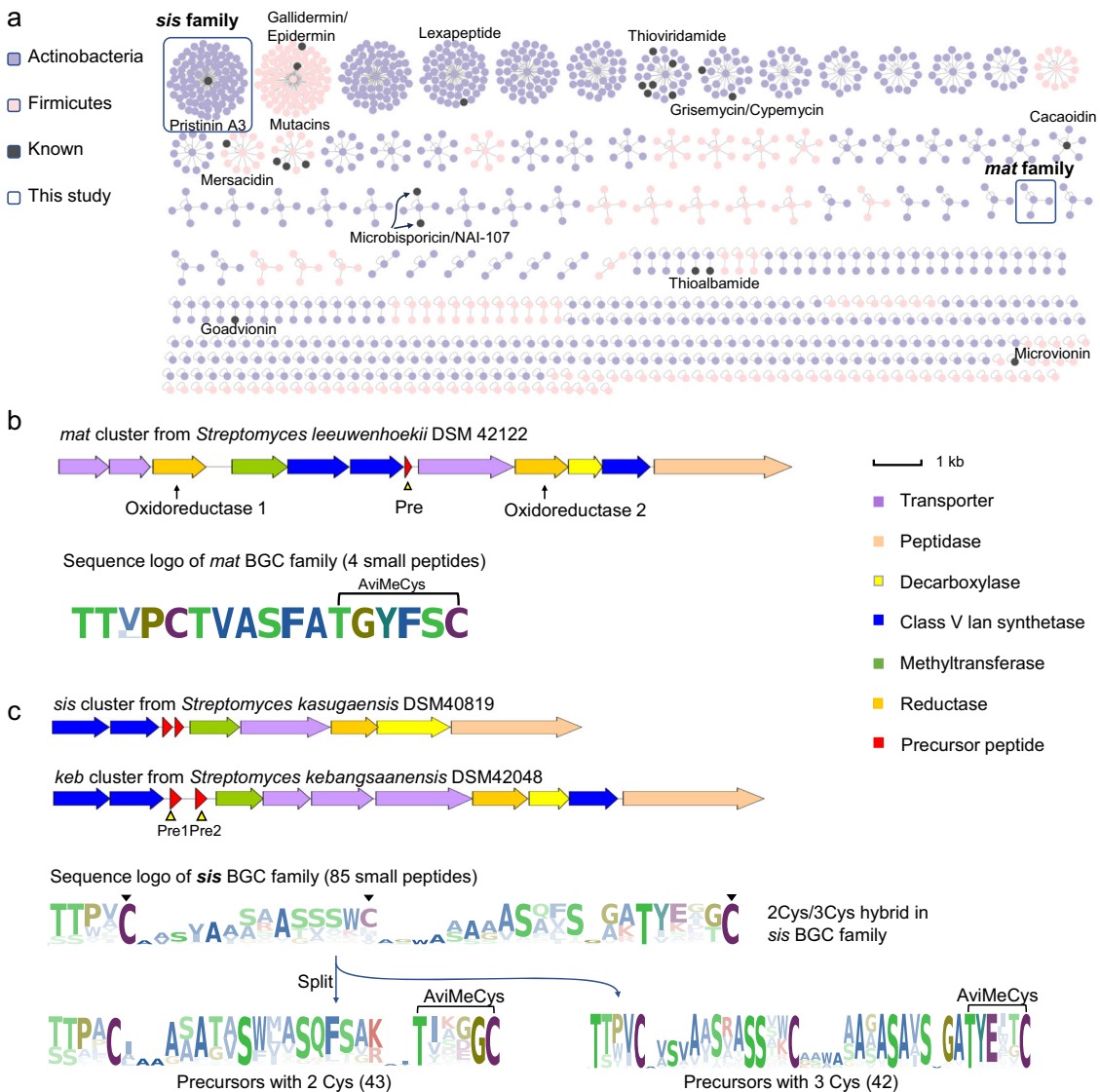

**Fig. 2 | SPECO-based genome mining of ACyP BGCs. a** A total of 1172 precursor peptides were identified by SSN analysis, which are categorized to 67 clusters. The sequence logo of each cluster was shown in Supplementary data 2. Previously characterized precursors are highlighted as black dots (microbisporicin and NAI-107 are synonyms and not distinct molecules). Two BGC families characterized in this study are boxed. **b** BGC architecture and core peptide sequence logo of the *mat* BGC family. Oxidoreductase and precursor were highlighted by arrow and triangle, respectively. The black connector denotes Avi(Me)Cys ring formation. **c** BGC architecture and core peptide sequence logo of the *sis* BGC family. Cys residues were highlighted by solid triangles.

## Linking Avi(Me)Cys biosynthetic gene clusters with peptide antibiotics

To discover new ACyPs from the BGCs of interest, we first focused on the *mat* cluster from *Streptomyces leeuwenhoekii* DSM42122 (Fig. 3a). This *mat* cluster contains several enzymes responsible for the formation of the (methyl)lanthionine ((Me)Lan) ring, including MatKYX enzymes, which exhibit similarities to the class V lanthionine (Lan) synthetase LxmKYX found in the lexapeptide BGC[33,46]. Other enzymes in the cluster include MatM, which functions as a methyltransferase, and MatT1-T3 and MatP, which act as transporters and peptidase, respectively. Additionally, MatR1 and MatR2 are putative oxidoreductases involved in the hydrogenation of Dha or Dhb[33,36]. These intricate modifications, encompassing decarboxylation, dehydration, thioether cyclization, hydrogenation, and methylation, give rise to a diverse range of possible monoisotopic masses for both intact and fragmented precursors (Fig. 3b, Supplementary data 4). Consequently, the identification of specific metabolites in the metabolic analysis of crude extracts can be challenging, as numerous mass-to-charge ratio

($m/z$) signals are observed, making it difficult to pinpoint the metabolites of interest.

To pinpoint the BGC-encoded ACyP from metabolites, the wild-type strain *Streptomyces leeuwenhoekii* DSM42122 was cultivated for 7 days and the high-resolution mass spectrometry (HRMS) data of the crude extract was collected. We then matched calculated mass data with experimentally detected metabolomic data to target natural products (Fig. 3b). This process involved enumerating the $m/z$ values of theoretical peptide fragments with various modifications, which were predicted based on post-translational modification rules (Fig. 3b). We then compared these theoretical values with the experimental HRMS data to find matches. For instance, UPLC-HRMS analysis of the culture extract showed typical peptidic signals and a mass matching pipeline revealed a hit of $[M + 2H]^{2+} = 817.8949$ (Fig. 3c, Supplementary note, Supplementary data 4). Further mass calculation suggested that the core region of the precursor peptide "TTVPCTVASFATGYFSC" (calculated $[M + 2H]^{2+} = 817.8965$, $\Delta$ppm = 2.0) underwent six dehydrations ($-6H_2O$), one decarboxylation ($-1CO_2H_2$), three

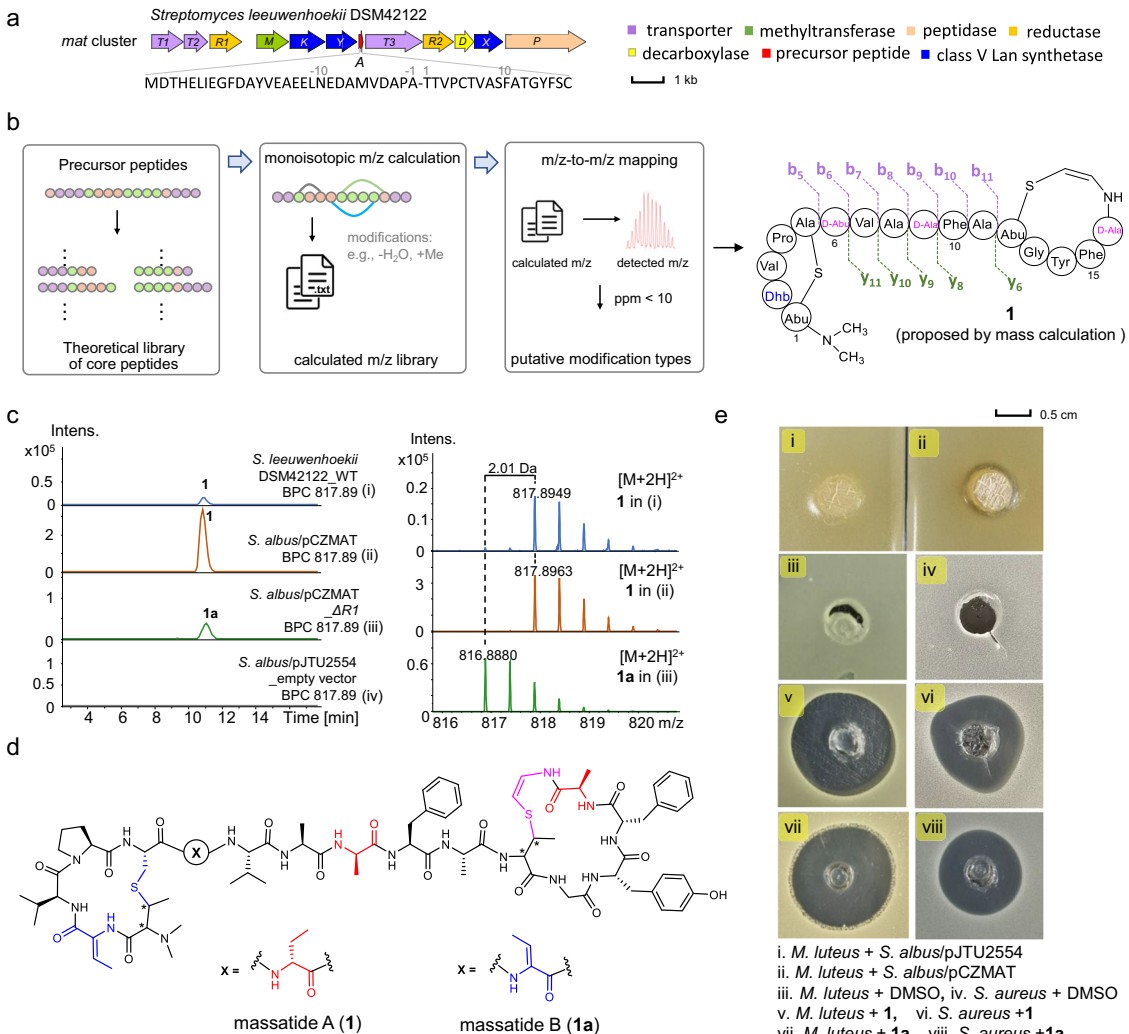

**Fig. 3 | Discovery and BGC characterization of massatides. a** The *mat* BGC and amino acid sequence of the precursor MatA. **b** Mass mapping pipeline and proposed structure of **1** based on tandem mass analysis. Calculation details are in the Supplementary note. **c** UPLC-HRMS analysis of *Streptomyces leeuwenhoekii* DSM42122 wild-type strain (i), *S. albus*/pCZMAT (ii, heterologous expression of *mat* BGC), *S. albus*/pCZMAT_*ΔR1* (iii, *matR1* deletion) and *S. albus*/pJTU2554 (iv, empty vector). **d** The structure of massatide. AviCys motifs are shown in pink. D-amino acids are shown in red. Other noncanonical amino acids are shown in blue. **e** Bioactivity of massatides against *S. aureus* and *M. luteus* (Fermentation culture was used in i and ii, and pure compounds or DMSO were used in iii−viii). For raw data, see Figure S3. Source data are provided in MassIVE (https://doi.org/10.25345/C5PC2TM3Q).

hydrogenations (+6H), and two methylations (+2CH₂). To solidify the association of the metabolite of $[M+2H]^{2+} = 817.8949$ (**1**) with the *mat* BGC, we cloned and heterologously expressed the entire *mat* BGC in *Streptomyces albus* J1074 (Table S3). UPLC-HRMS analysis revealed a new peak with $[M+2H]^{2+} = 817.8963$ in the *S. albus*/pCZMAT by comparison with the host strain harboring the empty pJTU2554 vector (Fig. 3c and S3). This result provided definitive evidence that the *mat* BGC encodes compound **1**, with MatA as the crucial precursor for structural elucidation. To delve deeper into the bioactivity of the *mat*-encoded product, we conducted an antibacterial assay utilizing a fermentation culture of the heterologous expression strain. The findings demonstrated that *S. albus*/pCZMAT exhibited inhibitory effects on the growth of *Micrococcus luteus* (Fig. 3e), establishing a direct correlation between the biosynthetic capacity and the antibiotic potential of the ACyPs.

With the confirmed core peptide sequence and calculated modifications in place, we aligned experimental tandem mass data with calculated monoisotopic mass data of fragment ions (Figure S4, Supplementary data 5). The results showed that experimental b5 to b11, y6, and y8 to y11 fragments with modifications matched well with

calculated ones. The matched modifications at ions b11 ($-4H_2O_- + 2CH_2 + 4H$) and y6 ($-2H_2O_- - 1CO_2H_2_ + 2H$) are consistent with the total of six dehydrations ($-6H_2O$) and C-terminal decarboxylation (Fig. 3b). Upon comparison of b5 to b6 and b8 to b9 ions, it was observed that two hydrogenated residues were localized at Thr6 and Ser9, respectively. The absence of fragment ions in the Thr1-Cys5 and Thr12-Cys17 motifs strongly suggests the formation of Thr1-to-Cys5 and Thr12-to-Cys17 linkages, with Ser16 being the only remaining residue that can be hydrogenated. It is noteworthy that hydrogenation during RiPP biosynthesis can potentially generate D-amino acids, as previously reported[33,47,48]. Based on the involvement of biosynthetic enzymes like methyltransferase and decarboxylase, we proposed a tentative structure for **1**, which we named massatide A (Fig. 3b).

To further validate the structure of compound **1**, we conducted a large-scale fermentation of the *S. albus*/pCZMAT strain, and 5 mg of **1** was purified for 1D and 2D nuclear magnetic resonance (NMR) analysis (Figures S5−S12, Table S4). In the ¹H and ¹³C spectra, characteristic signals of the Phe and Tyr side chains and the modified residues aminobutyric acid (Abu), Dhb, and AviMeCys were identified. The C-S formation in the AviMeCys motif was supported by key HMBC

correlations from Cys17*-Hβ (5.29 ppm, d, * denotes modification) to Thr12*-Cβ (45.3 ppm) and from Thr12*-Hβ (3.23 ppm) to Cys17*-Cβ (105.2 ppm) (Table S4 and Figure S12). A Z-geometry of the double bond in the AviMeCys residue was determined based on the corresponding $^3J_{H,H}$ value of 8.5 Hz (Table S4). Additionally, an N,N-dimethyl MeLan was observed based on HSQC and HMBC spectra, which is rarely reported except for a lanthidin RiPP[36]. To confirm absolute configurations for all the amino acid residues, we conducted advanced Marfey's analysis[49] for 1 using L/D-FDLA (5-fluoro-2,4-dini-trophenyl)-L/D-(leucinamide). The results showed the presence of both L- and D-Ala, with D-Abu also observed (Table S8). Further investigation showed that Ala8 and Ala11 were genetically encoded and in the L-configurations. Based on these results, we proposed that Thr6, Ser9, and Ser16 were converted to D-Abu and D-Ala in 1, while the remaining unmodified residues all existed as L-configuration. To the best of our knowledge, massatide A (1) represents the first Avi(Me)Cys-containing peptide with D-amino acids in the C-terminal AviMeCys ring.

Most previously reported lanthipeptide BGCs have only one reductase for Dha/Dhb hydrogenation[33,47,48]. However, in the mat BGC, both matR1 and matR2 encode LLM class oxidoreductases containing the conserved F420 binding domain, which belong to the LanJ$_C$ family of lanthipeptide reductases[33]. To determine the essentiality of both MatR1 and MatR2, we constructed a single gene knockout strain S. albus/pCZMAT_ΔR1. UPLC-HRMS data showed that compound 1a ($[M + 2H]^{2+}$ = 816.8880) from the knockout strain, named massatide B, was 2 Da lighter than 1 (Fig. 3c). Tandem mass analysis suggested that Thr6 was converted to Dhb in massatide B (1a) instead of D-Abu in 1 (Fig. 3d and S13). Thus, we confirmed that in the mat BGC, MatR1 catalyzes D-Abu6 formation, while MatR2 may be responsible for both D-Ala9 and D-Ala16 formation.

## Two crosslinking patterns catalyzed by one Avi(Me)Cys biosynthetic machinery

We next investigated the sis family, which is anticipated to produce ACyPs containing 2 and 3 thioether rings. This is due to the existence of precursor peptides containing 2 and 3 cysteine residues. We examined a sis BGC from Streptomyces kasugaensis DSM40819, which belongs to class V lanthipeptides with methylation and AviMeCys motif (Fig. 4a, Table S3). Two putative precursors, namely SisA1 and SisA2, contain three and two cysteines, respectively. This composition of cysteines suggests that SisA1 can potentially form a three-ring peptide, while SisA2 may give rise to a two-ring peptide. Utilizing the same PTM rule-guided metabolomic analysis workflow mentioned above, we pinpointed two putative peptide ion peaks corresponding to the precursors SisA1 (compound 2, observed $[M + 3H]^{3+}$ = 1085.2083, calculated $[M + 3H]^{3+}$ = 1085.2083, Δppm = 0.1) and SisA2 (compound 3, observed $[M + 3H]^{3+}$ = 816.7481, calculated $[M + 3H]^{3+}$ = 816.7487, Δppm = 0.7), respectively (Figure S14). The modifications of SisA1 occurred in the C-terminal 34-mer core region "TTYI-CASVAISRTSSVKCSAAASAISGATYEWTC", including eleven dehydrations (−11H$_2$O), one decarboxylation (−1CO$_2$H$_2$), five hydrogenations (+10H), and two methylations (+2CH$_2$), resulting in the production of 2 (Supplementary data 6). Similarly, modifications of the SisA2 core peptide "STPACGAATVSWIVSQFSAKTVKDGC" involved seven dehydrations (−7H$_2$O), one decarboxylation (−1CO$_2$H$_2$), and three hydrogenations (+6H), leading to the formation of 3 (Supplementary data 6). To validate the ribosomal origins of compounds 2 and 3, we cloned and heterologously expressed the entire sis BGC in S. albus (Table S3). Comparative UPLC-HRMS analysis revealed two new peaks in S. albus/pCZSIS, i.e., $[M + 3H]^{3+}$ = 1085.2076 and $[M + 3H]^{3+}$ = 816.7487, which are respectively identical to 2 and 3 from the wild-type strain (Figure S14). The results confirmed that 2 and 3, named sistertide A1 and A2, were indeed encoded by the sis BGC. In the antibacterial assay using a fermentation culture of the heterologous expression strain, S.

albus/pCZSIS exhibited growth inhibition against M. luteus, confirming the antibiotic potential of these sistertides (Fig. 4a).

Tandem mass analysis of sistertide A1 (2) fragments revealed N-terminal methylation and C-terminal decarboxylation (Fig. 4a and S15). Furthermore, the analysis indicated that Ser7, Ser11, Ser19, Ser23, and Ser26 respectively underwent a 16 Da mass loss, suggesting that they were first dehydrated (−H$_2$O) and then reduced (+2H). Notably, no fragment ions were observed around three cysteines, which implied that Cys5, Cys18, and Cys34 might cyclize with Thr1, Ser14, and Thr29 (Fig. 4a and S15). We next performed a large-scale fermentation of S. albus/pCZSIS to determine the cross-linking patterns and obtained 4 mg pure sistertide A1 (2) for NMR analysis (Figures S16–S23, Table S5). Characteristic signals of the side chains of Tyr, Trp, Thr, and modified residues Dha and Dhb were observed in 2. NMR analysis also showed that 2 contains an AviMeCys scaffold with a Z-geometry ($^3J_{H,H}$ 8.4 Hz) at the double bond, which was derived from the Thr29/Cys34 motif of the core peptide (Table S5). The presence of N,N-dimethyl MeLan was confirmed by key HMBC correlations (Figures S20 and S23). However, due to its low yield, we could not collect enough sistertide A2 (3) for NMR analysis. Mass calculation and tandem mass data of 3 suggested that the core peptide of SisA2 underwent dehydration, decarboxylation, hydrogenation, and ring formations, which led to the formation of one lanthionine ring and one AviMeCys ring (Fig. 4a and S24).

Sistertide A1 is methylated, while A2 lacks this modification despite being modified by the same enzymes. This suggests that the methyltransferase in the sis BGC may exhibit selectivity in substrate recognition. The presence of a putative oxidoreductase SisR in this BGC suggests possible D-amino acid formation in sistertides. Advanced Marfey's analysis[49] for 2 revealed the presence of both L- and D-Ala in the structure (Table S9), with the D-Ala residues likely converted from Ser7, 11, 19, 23, and 26 in the precursor. The remaining unmodified residues were L-configurated. Similarly, for sistertide A2 (3), Ala11, Ala15, and Ala18 likely resulted from dehydration and reduction of Ser residues in the precursor, suggesting that they may also be D-Ala in the mature compound based on biosynthetic logic.

Additionally, within the sis family, we identified two other ACyPs encoded by a sis BGC homolog, the keb cluster from Streptomyces kebangsaanensis DSM42048 (Fig. 4b). Our initial analysis of the wild-type DSM42048 using the workflow revealed no discernible target peaks from the fermentation. Therefore, we hypothesized that the BGC might be either inactive or expressed at a very low level in the native host. Subsequently, we tried to heterologously express the entire keb BGC in Streptomyces albus, which resulted in the production of two new compounds, 4 (named kebanetide A1, $[M + 3H]^{3+}$ = m/z 921.4331) and 5 (named kebanetide A2, $[M + 3H]^{3+}$ = m/z 793.7395) (Fig. 4c and S25). Using mass calculation, tandem mass spectrum analysis (Supplementary data 7), $^1$H NMR spectroscopy, and advanced Marfey's analysis, we determined that kebanetide A1 (4) features one C-terminal AviMeCys ring and two thioether crosslinks, while kebanetide A2 (5) resembles sistertide A2 (3), containing a C-terminal AviMeCys ring and an N-terminal thioether ring (Fig. 4d and S26–S33, Table S6–S7 and S10–S11).

Although the leader peptide remains conserved across multiple precursors within the sis family, significant variations are observed in the core regions of precursors A1 and A2, particularly in terms of their length and the number of essential cysteine residues involved in ring formation. Notably, the AviMeCys ring of sistertide A1 is characterized by the presence of two bulky hydrophobic aromatic side chains (Tyr30 and Trp32) and one negatively charged residue (Glu31), while sistertide A2 features a positively charged lysine residue (Lys23). These substantial disparities in the core peptide imply that the tailoring enzymes within the sis BGC family display broad substrate selectivity. It is worth noting that the discovery of 42 BGCs in this family indicates that the "one tailoring enzyme system constructs multiple crosslinking

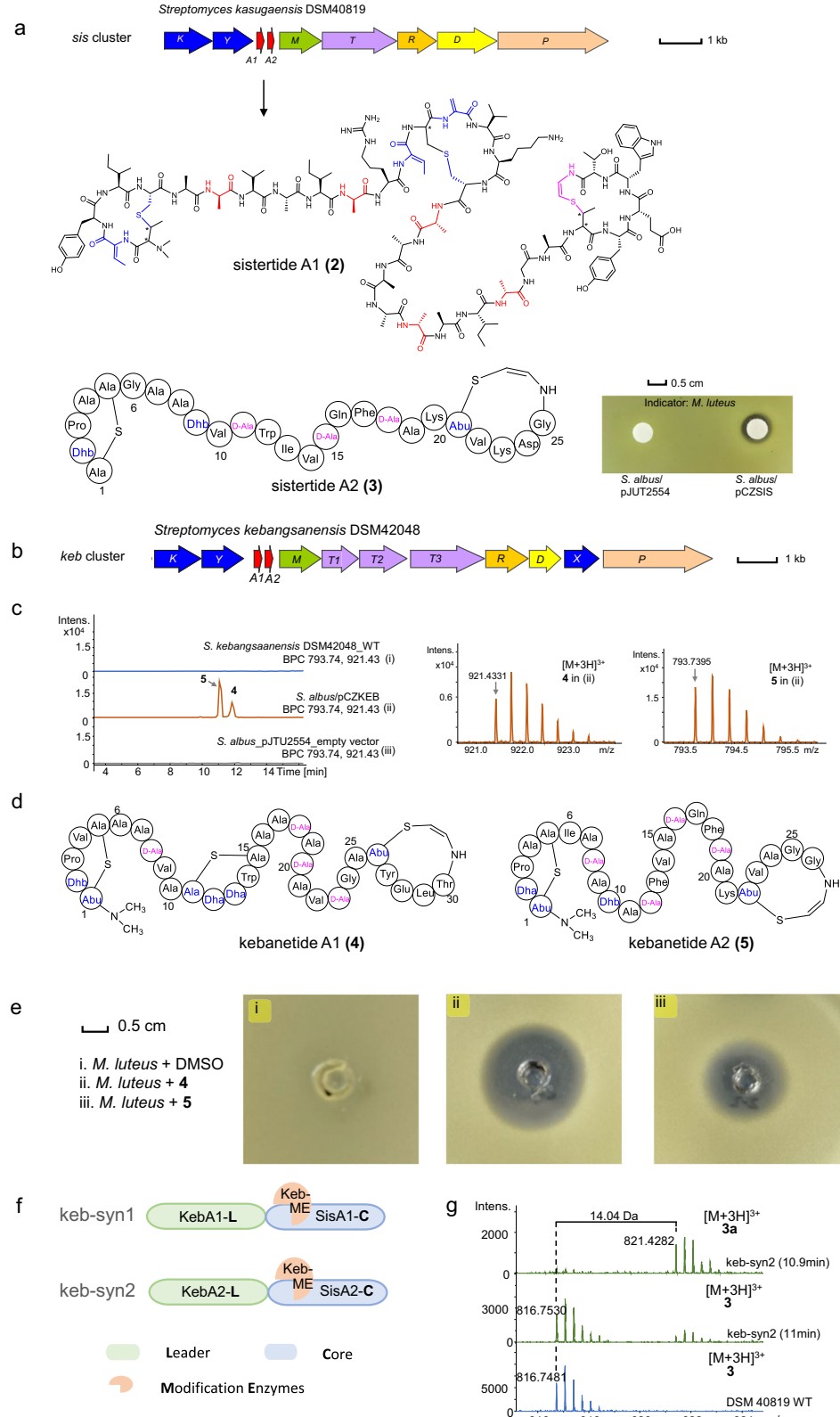

**Fig. 4 | Discovery and characterization of the *sis* BGC family. a** The *sis* BGC, structure of sistertide A1 and suggested structure of sistertide A2, and antibacterial potential of *S. albus*/pCZSIS crude extract. The AviMeCys motif is shown in pink. D-amino acids are shown in red. Other noncanonical amino acid are shown in blue. **b** The *keb* BGC. **c** UPLC-HRMS analysis of *Streptomyces kebangsaanensis* DSM42048 wild-type strain (i), *S. albus*/pCZKEB (ii, heterologous expression of *keb* BGC) and *S.*

*albus*/pJTU2554 (iii, empty vector). **d** Suggested structures of kebanetides. **e** Bioactivity of kebanetides against *M. luteus*. **f** Chimeric precursor with the maturase from *keb* BGC. **g** UPLC-HRMS analysis of new-to-nature analog from construct keb-syn2. Source data are provided in MassIVE (https://doi.org/10.25345/C5PC2TM3Q).

**Table 1 | MIC values of massatides, sistertide A1 and kebanetides (μg/mL)**

| Strains | | Massatide A (1) | Massatide B (1a) | Sistertide A1 (2) | Kebanetide A1 (4) | kebanetide A2 (5) | Nisin | Vancomycin |
|---|---|---|---|---|---|---|---|---|
| G+ | *Staphylococcus aureus* ATCC25923 | 0.5 | 1 | >64 | 64 | >64 | 64 | 1 |
| | *Micrococcus luteus* DSM1790 | 0.06 | 0.25 | >64 | 8 | 32 | 0.5 | 0.125 |
| | *Bacillus subtilis* 168 | 2 | 4 | >64 | 8 | 32–64 | 32 | 0.125 |
| | *Enterococcus faecalis* OG1RF | 4 | 4 | >64 | 16 | 16 | >64 | 2 |
| | *Enterococcus faecium* MCC2763 | 2 | 4 | >64 | 32 | 32 | >64 | 1 |
| G− | *Escherichia coli* DH5α | >128 | >64 | >64 | >64 | / | / | / |
| | *Pseudomonas aeruginosa* PAO1 | >128 | >64 | >64 | >64 | / | / | / |

patterns" are widespread among actinobacteria. RiPP biosynthetic cassettes with multiple precursors were found in two-component class II lantibiotics, such as lichenicidin[50] and haloduracin[51]. However, in contrast to the *sis* family, multiple LanM enzymes are encoded in these two-component lantibiotics BGCs to catalyze precursors in a one-LanM-one-precursor manner. The ribosomal peptide maturases in the *sis* family resemble promiscuous lanthipeptide synthetase ProcM, which has been widely used to generate cyclic peptide libraries[52].

To explore the catalytic potential of the *sis* family maturase, we conducted a leader-core mixing and matching assay. We designed two chimeric precursor peptides (Syn1 and Syn2), each combining the KebA leader with the SisA core region, to investigate the substrate tolerance in ACyP biosynthesis (Fig. 4f and S34). Co-expression of these two chimeric precursors with the maturases from the *keb* BGC (constructs keb-syn1 and keb-syn2) led to the production of compound **2a**, sistertide A2 (**3**) and **3a** (Fig. 4g and S35–36). Tandem mass analysis supported that **3a** is monomethylated **3** at the N-terminus (Figure S36), indicating the *keb* maturase can not only modify the core peptide of SisA2, but also generate new-to-nature analogs via methylation. HRMS data revealed that **2a** is a new derivative compared to sistertide A1 (**2**). Further tandem mass spectrometry analysis indicated that the new modification was localized at the N-terminus (Figure S35). However, no new peaks were observed when chimeric precursors Syn1 and Syn2 were co-expressed with the *sis* maturases (constructs sis-syn1 and sis-syn2, Figure S34), implying that leader-maturase mismatching hinders core peptide modification. Taken together, these findings suggest that the tailoring enzymes from the *keb* BGC exhibited selectivity for leader peptide but tolerance for core peptide.

## Potent antibacterial activity of isolated ACyPs

We next employed the Kirby-Bauer assay to evaluate the antibacterial properties of isolated ACyPs from the *mat* and *sis* families. Massatides A (**1**) and B (**1a**) showed strong inhibitory activity against *Staphylococcus aureus* and *Micrococcus luteus* (Fig. 3e). Similarly, kebanetides A1 (**4**) and A2 (**5**) exhibited growth inhibition against *M. luteus*, as shown in Fig. 4e. Moreover, we determined the minimal inhibitory concentrations (MICs) of massatide A, massatide B, sistertide A1, kebanetide A1, and A2 against seven bacterial strains to assess their bioactivity, as detailed in Table 1. Massatide A demonstrated a broad-spectrum antibacterial activity against gram-positive bacteria, with lower MICs than nisin and comparable activity to vancomycin. Its antimicrobial activities against *S. aureus* and *M. luteus* even surpassed those of vancomycin, highlighting its potential as a candidate for gram-positive antibiotics. The activity of massatide B was similar to that of massatide A, with only a slightly larger MIC, indicating that the presence of D-Abu6 has a minimal contribution to the antibacterial activity. Kebanetide A1 and A2 demonstrated weaker bioactivity than massatides but performed better than nisin against *E. faecalis* and *E. faecium*. The crude extract of the sistertides producer strain exhibited a zone of inhibition (Fig. 4a), even though sistertide A1 alone did not display significant inhibitory activity. This suggests that the combined action of sistertide A1 and A2 is necessary for their inhibitory activity.

No bioactivity was detected against gram-negative bacteria, indicating that these ACyPs specifically target gram-positive bacteria.

## Efficacy of massatide A in vitro and in vivo

In light of the promising antibacterial potential of massatide A, we conducted extensive research to investigate its antibacterial efficacy both in vitro and in vivo. The growth kinetics and time-dependent killing assay revealed that massatide A was bactericidal against *S. aureus* and was superior to vancomycin, the first line of defense antibiotic, in killing early exponential phase populations (Fig. 5a, c). We noticed that massatide A did not result in lysis of the cell culture, which is the same as vancomycin (Fig. 5b). In vitro stability assay showed that massatide A is resistant to trypsin, chymotrypsin, and many other proteases (Figure S37) and highly stable under high temperatures (Figure S38). The subsequent antibacterial assays yielded remarkable results, demonstrating that massatide A exhibited excellent activity against a range of clinically isolated resistant pathogens, as outlined in Table 2. Its potency against tested gram-positive pathogens, including vancomycin intermediate-resistant strains, remained below 4 μg/mL. Particularly noteworthy was its exceptional performance against linezolid-resistant *S. aureus* 20-1 and methicillin-resistant *S. aureus* ATCC43300, with a MIC of only 0.25 μg/mL. Over a span of 25 days, serial passaging in subinhibitory concentrations of massatide A resulted in only a slight increase in the MIC to *S. aureus*, from 0.5 to 4 μg/mL. No significant change in MIC was observed during the subsequent 19 days after day 6, suggesting that *S. aureus* did not develop significant resistance to this compound. This finding further supports the potential of massatide A as an effective antibacterial agent.

Massatide A exhibits cytotoxicity against Hela cells, with an IC50 value of 136.3 μg/mL, while massatide B did not exhibit any significant cytotoxicity against Hela cells. However, both massatide A and B demonstrated moderate toxicity against the human cell line Hek293T (Figure S39). To further assess its safety profile, an acute toxicity study was conducted in mice, where massatide A was administered at doses ranging from 10-50 mg/kg. After 5 days of injection, all treated mice survived without any signs of acute toxicity (Fig. 5e). These findings suggest that massatide A has a favorable safety profile, highlighting its potential as a relatively safe and effective antibacterial agent.

Expanding upon its potent antibacterial activity, low in vivo toxicity, and relatively low risk of resistance, we conducted further evaluations to assess the efficacy of massatide A in a mouse septicemia model. Mice were intraperitoneally infected with $7.2 \times 10^9$ c.f.u. of MRSA ATCC43300, and lethality was observed after 7 days of infection (Fig. 5f). Remarkably, the survival rate of mice significantly improved to 80% at a single dose of 10 mg/kg of massatide A, and further increased to 100% at a higher dosage of 50 mg/kg, in contrast to the control group with no survivors. The findings strongly suggest the efficacy of massatide A in vivo, showcasing its potential as a therapeutic agent against gram-positive pathogens. With its remarkable efficacy and favorable safety profile, massatide A is a candidate for further research and development in antibacterial treatment. These

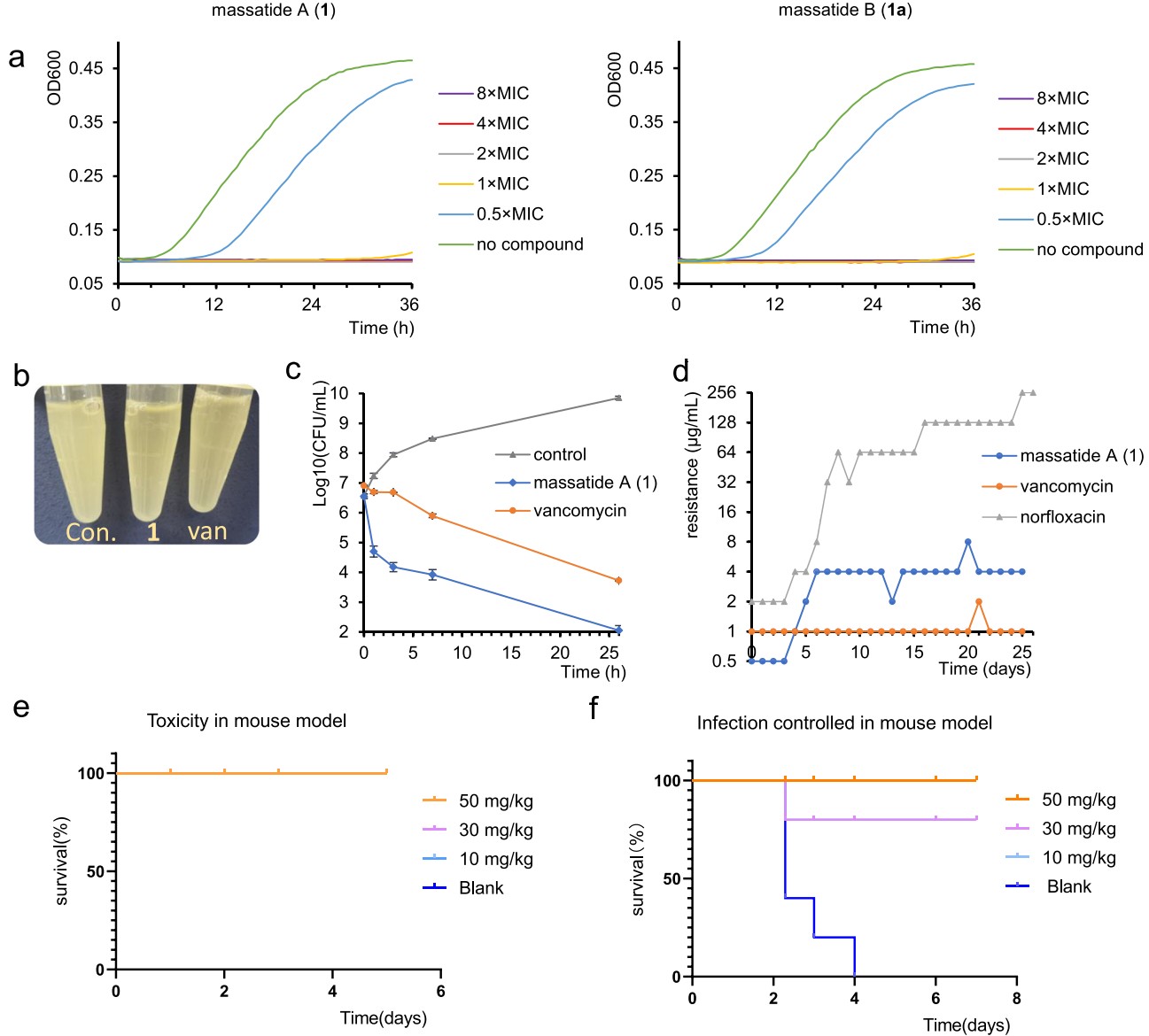

**Fig. 5 | Bioactivity of massatide. a** The growth kinetics of *S. aureus* with massatide A and massatide B. **b** Massatide A did not resulted in lysis. (Con. is the control with no compound.) **c** Time-dependent killing of *S. aureus* (10×MIC vancomycin, 10×MIC massatide A). Time points are graphed as the mean c.f.u. ± s.d (*n* = 3). The experiment was performed in biological triplicate. **d** Resistance acquisition during serial passaging in the presence of sub-MIC levels. Vancomycin and norfloxacin were used as control. **e** The survival rate in acute toxicity study in vivo (Blank and all the experimental groups are overlapped). **f** The survival rate in septicemia model using MRSA treated with massatide A (10 mg/kg and 30 mg/kg groups are overlapped). The relevant Source data for Fig. 5a, c, e, f are provided as a Source Data file.

results underscore the tremendous significance of ACyP macrocyclic peptides in the discovery of new antibiotics, particularly in addressing the critical need for effective antibacterial agents to combat drug-resistant bacteria.

## Discussion

Bacterial ribosomal peptides have long been recognized as a valuable source for antibiotic discovery[26]. However, the task of identifying new antibiotic peptides remains formidable due to high rates of rediscovery. Recent advancements in bacterial genome and meta-genome sequencing have revealed a previously untapped reservoir of BGCs[53]. Nevertheless, the challenge lies in prioritizing BGCs of interest and effectively connecting produced metabolites to their respective BGCs[16,26–29]. In this study, we introduce a comprehensive approach that integrates rule-based genome mining, PTM rule-guided metabolomic analysis, and heterologous expression as a platform to address these challenges. Based on the biosynthetic rule of ACyPs, our rule-based

omics approach effectively connects the potential of biosynthesis to actual chemical compounds and eventually to their bioactive potential. Our work demonstrates progress toward targeted genome mining for antibiotic discovery and highlights the utility of multiple computational strategies to circumvent rediscovery issues and accelerate drug discovery[26,28,29]. We successfully discovered one ACyP, massatide A, that exhibited potent antibacterial activity against a wide range of gram-positive pathogens. However, it is worth noting that our approach may have overlooked potential novel compounds with unanticipated modifications that were not predictable by rule-based metabolomic analysis. The presence of unknown modification enzymes associated with BGCs suggests the need for enhanced strategies in enzyme function annotation and metabolomics analysis in rule-based omics mining. This includes predicting enzymatic modifications and mapping their corresponding masses (both MS1 and MS/MS pattern), which can be valuable in addressing such scenarios. Nonetheless, our study demonstrates the potential of a

**Table 2 | Biological activity of massatide A against clinic isolated resistant strains (µg/mL)**

| Strains | | Massatide A(1) | Levofloxacin |
|---|---|---|---|
| G+ | MRSA 15-1 | 2 | 16 |
| | LRSA 20-1 | 0.25 | 32 |
| | MRSA ATCC33591 | 1 | 0.25 |
| | MRSA ATCC43300 | 0.25 | 0.25 |
| | VMSA ATCC700788 | 4 | 32 |
| | VMSA ATCC700699 | 4 | 64 |
| | *Enterococcus faecalis* 19-1 | 4 | 1 |
| | LREfa 20-1 | 4 | 32 |
| | VREfa ATCC51575 | 4 | 0.5 |
| G− | *Acinetobacter baumannii* 20-1 | 64 | 8 |
| | *Acinetobacter baumannii* 20-2 | 64 | 0.06 |
| | *Klebsiella pneumoniae* ATCC700603 | >64 | 0.5 |
| | *Klebsiella pneumoniae* 15-1 | >64 | 0.25 |

MRSA Methicillin-resistant *Staphylococcus aureus*, LRSA Linezolid-resistant *Staphylococcus aureus*, VMSA Vancomycin-intermediate *Staphylococcus aureus*, LREfa Linezolid-resistant *Enterococcus faecalis*, VREfa Vancomycin-resistant *Enterococcus faecalis*.

comprehensive approach to discovering novel antibiotic peptides from the underexplored realm of RiPP families.

Flavoproteins, a family of essential enzymes in ACyP biosynthesis, are widely spread in bacterial genomes. Using the SPECO genome mining pipeline, we identified 1172 BGCs within actinobacteria and firmicutes genomes, encompassing 67 distinct ACyP families, which hold great potential for antibiotic discovery. These diverse ACyP BGCs, in combination with the widely distributed flavoprotein in bacteria, represent an underexplored source for exploring the chemical diversity of Avi(Me)Cys-containing RiPPs. Notably, this highlights their potential as a valuable yet untapped resource for genomics-guided peptide antibiotic discovery. For instance, in addition to known ACyP families, a recent finding has revealed that flavoproteins could collaborate with radical *S*-adenosylmethionine enzymes to catalyze the synthesis of a new unsaturated thioether residue, *S*-[2-aminovinyl]-3-carbamoylcysteine (AviCamCys)[54]. Whether this new structural moiety can bestow unique biological activity remains to be explored. The discovery of flavoproteins across various RiPP families emphasizes the potential for exploring the untapped sequence space of flavoproteins to discover novel RiPPs.

Our rule-based omics mining strategies have led to the identification of massatide A, a peptide antibiotic effective both in vitro and in vivo. These findings highlight the significant potential of ACyPs as valuable candidates for therapeutic use against gram-positive pathogens. In comparison with gallidermin[55], NAI-107[31], and NVB302[56], which have advanced to preclinical development stages, massatide A features relatively smaller molecular weight and more concise crosslinks (Figure S40). Its antibacterial activity against gram-positive bacteria was superior to that of gallidermin and NVB302[55,56]. Avi(Me)Cys-containing lanthipeptides are reported to target lipid II, an essential precursor in cell wall biosynthesis[30]. One characterized ACyP is cacaoidin, which displays a dual mechanism of action by binding to lipid II and interacting with cell wall transglycosylase[57]. Massatide A, which shares structural similarities with cacaoidin, may potentially target lipid II or certain membrane proteins involved in peptidoglycan biosynthesis. However, a comprehensive study is necessary to fully uncover the detailed mode of action of massatide A, which will be the primary focus of our ongoing research. In terms of the resistance risk, it is worth noting that vancomycin-resistant strains have been reported since 1986[10], despite the absence of a shift in our experiment (Fig. 5d). Therefore, it is crucial to take into account the potential development of resistance to massatide A with prolonged usage, especially if modifications occur to the structure of its putative target, such as lipid II, in bacteria. Furthermore, there is an ongoing need for further structural enhancements to refine its drug-like characteristics, including safety profiles, especially regarding its potential toxicity in human kidney cells. Interestingly, the substrate-promiscuous maturases from the *sis* family offers a possibility for bioengineering new-to-nature analogs with potential antibiotic activity. Thus, we anticipate that by harnessing the flexibility of ACyP biosynthesis, we will pave the way for creating potent antibiotics characterized by simpler structures, higher efficacy, and lower toxicity. Ultimately, this will lead to the introduction of safer and more effective antibacterial agents in the future.

It is noteworthy that previous studies in natural product biosynthesis have accumulated a wealth of biosynthetic rules. We believe that now is the opportune time to harness this accumulated knowledge to guide the design of new genome mining strategies. Our research clearly demonstrates the effectiveness of rule-based omics analysis in prioritizing biosynthetic gene clusters (BGCs) capable of producing natural products with desired functional moiety(s) and linking them to antibiotic candidates. We envision that our efforts in genome mining, biosynthesis, and bioactivity study of ACyPs will contribute to the discovery of new RiPP natural products, thereby expanding the repertoire of compounds available for antibiotic development. This, in turn, holds the potential to address the urgent need for novel antibiotics to combat the rise of antibiotic-resistant bacteria.

## Methods
All animal experiments in this study were approved by the Animal Care and Use Committee of the Institute of Materia Medica, Chinese Academy of Medical Sciences, and Peking Union Medical College (Beijing, China). Strains, biochemical reagents, and primers used in this study are listed in the Supplementary Information.

### Bioinformatic analysis
Bacterial genomes were downloaded from the NCBI database (date: 2021-08-11). Small peptide and flavoprotein pairs were fetched based on the biosynthetic logic using the SPECO[24]. Specifically, the calculation is based on (i) the small peptide length should be between 20 and 100 amino acids; (ii) small peptide should end with cysteine and the C-terminus should feature T/S-Xn-C, where T and S indicate threonine and serine respectively; Xn ($2 \le n \le 6$) is the number of gapped residues between Thr/Ser and Cys; (iii) PF02441 defined flavoprotein; and (iv) the distance between small peptide and flavoprotein should no longer than 15,000 base pairs. SSN analysis of small peptides was performed by using MMseqs2[58] tool with the following parameters: mmseqs --easy-cluster --min-seq-id 0.6 --seq-id-mode 1 --cluster-mode 1 --similarity-type 2 --single-step-clustering --dbtype 1. Sequence logos were generated by using following method[24]: (1) the cluster files generated using MMseqs2 were first parsed to generate individual fasta file using script (https://github.com/yxllab-hku/ACyPs_code/tree/main/cluser_to_logo/cluster_parser.py) and (2) for each generated fasta file, the sequence logo will then be generated using the script (https://github.com/yxllab-hku/ACyPs_code/tree/main/cluser_to_logo/makelogo.py).

### Fermentation and chemical extraction of actinobacteria
The actinobacteria were spread on agar plates for sporulation and growth. All the wild-type strains used optimized GYM medium (glucose 4 g/L, yeast extract 4 g/L, malt extract 10 g/L, peptone 1 g/L, NaCl 2 g/L, CaCO$_3$ 2 g/L, pH 7.2, agar 2%). *S. albus* used MS medium (soybean flour 25 g/L, mannitol 20 g/L, pH 7.2–7.4, agar 2%). After cultivating for 3 days, strains were transferred to a fermentation medium (GYM, with

2% agar or R5A (sucrose 100 g/L, $K_2SO_4$ 0.25 g/L, $MgCl_2$ 10.12 g/L, glucose 10 g/L, yeast extract 5 g/L, MOPS 21 g/L, pH 6.85, agar 2%, 200 μL trace element solution*)) for cultivating another 7 days at 30 °C. The culture was extracted three times with methanol. The methanol extract was dried under rotary evaporation and dissolved in water. Next, we used an equal volume of n-butanol to extract three times. The organic phase was dried under rotary evaporation and then dissolved in methanol for MS analysis or isolation.

(* 100 mL trace element solution contains $ZnCl_2$ 40 mg, $FeCl_3\cdot6H_2O$ 200 mg, $CuCl_2\cdot2H_2O$ 10 mg, $MnCl_2\cdot4H_2O$ 10 mg, $Na_2B_4O_7\cdot10H_2O$ 10 mg, $(NH_4)_6Mo_7O_{24}\cdot4H_2O$ 10 mg).

### Construction of heterologous expression strains

The *mat* BGC was first divided into three fragments for PCR amplification from the DSM42122 genome and then directly cloned into the pJTU2554 vector using HiFi DNA assembly, yielding the recombinant plasmid pCZMAT. The resulting plasmid was transferred into *S. albus* J1074 via bi-parental conjugation[59] using ET12567/pUZ8002/ pCZMAT (donor strain) and *S. albus* J1074 (recipient strain) spores. Colonies that could grow on an MS plate with apramycin and nalidixic acid at 30 °C and confirmed by PCR were identified as target recombinant strain *S. albus*/pCZMAT. The *sis* BGC and *keb* BGC were amplified from DSM40819 and DSM42048 genomes, respectively. The construction of *S. albus*/pCZSIS and *S. albus*/pCZKEB are the same as above. The fermentation of recombinant strains using the same protocol as wild-type strains.

### Determination of antibacterial activity and MIC value

The stock solution of massatide A (**1**) and B (**1a**) 3.2 mg/mL were prepared in DMSO. 40 μL overnight culture of *Micrococcus luteus* DSM1790 was added to 20 mL soft Luria-Bertani agar (1% agar), and then poured into a plate. 10 μL tip was used to make two wells on the plate. 10 μL massatide A (**1**) or B (**1a**) solution was added in one of the wells, while in the other well, 10 μL DMSO was added as a negative control. The plates were incubated at 30 °C for 20 h to observe the zone of inhibition. *Staphylococcus aureus* ATCC25923 was also used to observe the inhibition zone using a similar method. The only difference is incubating at 37 °C. Stock solutions of sistertide A1, kebanetide A1 and A2 were 3.2 mg/mL, 1.6 mg/mL and 1.6 mg/mL, respectively. The method to observe the activity of these compounds individually was the same as that of massatide.

For MIC determination, *Staphylococcus aureus* ATCC25923, *Bacillus subtilis* 168, *Micrococcus luteus* DSM1790, *Enterococcus faecalis* OG1RF, *Enterococcus faecium* MCC2763, *Escherichia coli* DH5α and *Pseudomonas aeruginosa* PAO1 were used as indicator strains. Nisin and vancomycin were used as positive controls. A dilution series of massatide A, massatide B, vancomycin (16, 8, 4, 2, 1, 0.5, 0.25, 0.125, 0.0625, 0.0313 μg/mL) and sistertide A1, kebanetide A1, kebanetide A2, nisin (64, 32, 16, 8, 4, 2, 1, 0.5, 0.25, 0.125 μg/mL) were prepared in 96-well plate from 1–10th column. The indicator strains were added to the compound-containing well. The indicator strain culture with no antibiotics was prepared in the 11th column. The 12th column was filled with 200 μL empty MHB medium. The 96-well plate was incubated at 37 °C for 20 h (30 °C for *M. luteus* and *B. subtilis*). The $OD_{600}$ was detected on MULTISKAN Sky (Thermo Scientific) and MIC values were determined. The MIC values of clinical isolates were determined by Sichuan Primed Shines Bio-tech Co., Ltd. (China) following similar protocol. Levofloxacin was used as a positive control.

### Animal studies

Animals were obtained from SPF (Beijing) Biotechnology Co., Ltd. (China). All animal experiments were approved by the Animal Care and Use Committee of the Institute of Materia Medica, Chinese Academy of Medical Sciences, and Peking Union Medical College (Beijing, China). Animal care and experimental procedures were conducted in accordance with the Beijing Administration Rule of Laboratory Animals. Mice were housed at 22 °C and relative humidity 50%, with a 12:12 h light:dark cycle.

### Mouse acute toxicity test

Massatide A was dissolved in saline to the required concentration for the experiments. Six-week-old specific pathogen-free male ICR mice weighing 18–20 g were randomly divided into 4 groups (4 per group) and administered in a single intraperitoneal injection. Group 1 animals received 0.9% saline only, while groups 2–4 were given massatide A at 10, 30, and 50 mg/kg, respectively. The animals were continuously observed for general behavioral changes for 5 days and survival rate was recorded.

### Mouse septicaemia model

Briefly, 6-week-old specific pathogen-free male ICR mice weighing 18–20 g were randomly divided into 4 groups (5 per group). Each mouse received an intraperitoneal injection on the left side of the abdomen with 0.4 ml, $1.8\times10^{10}$ c.f.u./mL MRSA ATCC43300 suspended in saline containing 5% yeast in each mouse. One hour after infection, the mice were injected intraperitoneally with saline (negative control), massatide A dissolved in saline at 10 mg/kg, 30 mg/kg and 50 mg/kg (experimental groups). The concentration of massatide A stock solution (in 5% DMSO, 5% Tween-80, 90% saline) is 2.5 mg/ml (for 50 mg/kg group), 1.5 mg/ml (for 30 mg/kg group) and 0.5 mg/ml (for 10 mg/kg group). We injected 0.4 ml solution to each mouse. Animal deaths were recorded for 7 days, and the survival rates were calculated to generate the mouse survival curve.

### Statistics & Reproducibility

No statistical method was used to predetermine the sample size. No data were excluded from the analyzes.

### Reporting summary

Further information on research design is available in the Nature Portfolio Reporting Summary linked to this article.

## Data availability

Data supporting the results of this study is available in the paper, supplementary information and supplementary data. Source data are provided with this paper. The HRMS data generated in this study have been deposited in MassIVE under accession code MSV000094648[60]. Source data are provided with this paper.

## Code availability

Data from mass calculation and mapping were analyzed using scripts written in the programming language Python. The code of the scripts is available from https://github.com/yxllab-hku/ACyPs_code. The source code used in the paper has also been assigned a citable DOI through Zenodo (https://doi.org/10.5281/zenodo.11099314)[61].

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

## Acknowledgements

This work is funded by the National Key Research and Development Program of China (2018YFA0903200) and the Research Grants Council of Hong Kong (27107320, 17115322, and 17102123) to Y.X. L., Shenzhen Bay Laboratory Funds (21230051 and SZBL2021080601006) and Guangdong Province's Pearl River Recruitment Program of Talents (2021QN02Y855) to X.T., and the CAMS Innovation Fund for Medical Sciences (2021-I2M-1-069) to S. W. The authors would like to thank Prof. Meifeng Tao for the gift of pJTU2554 vector and Dr. Prasanna Neelakantan for the gift of indicator strains. We also thank Yi-Man Eva Fung, Jo Yip and Bonnie Yan for their help in MS and NMR analysis.

## Author contributions

Y.X. L., X. T., W. Z, Z. C. and B.B. H. designed the project and prepared the manuscript. Z. C. performed experiments and MS data analysis. B.B. H. and Z. Z performed bioinformatic analysis. K. L. conducted the animal experiments under the supervision of W. Z., and S. W. Y. S. performed mammalian cytotoxicity. Y. G. contributed to the mass mapping pipeline. H. L. and R. L. contributed to the NMR data analysis. H. Z. contributed to the manuscript proofreading. Y.X. L. supervised the project.

## Competing interests

The authors declare no competing interests.
