## [Peer Review File · Nature Communications]

REVIEWER COMMENTS

Reviewer #1 (Remarks to the Author):

In this study, Cheng et al describe the discovery of several new ribosomally synthesized and post-translationally modified peptides (RiPPs) with antibiotic properties. The authors apply their recently described SPECO approach (Short Peptide and Enzyme Co-Occurrence analysis pipeline) to mine publicly available genome data for short peptide-encoding genes that co-occur with a specific flavoprotein. This enzyme is assumed to introduce a aminovinyl-(methyl-)cysteine moiety, which is a modification found in several families of RiPPs, most prominently in the linaridin class of antibiotics. The authors use their bioinformatics pipeline to detect small putative precursor peptides co-occurring in the vicinity of such flavoproteins, cluster them based on sequence similarity of the precursor peptides, and prioritize candidates from two promising clusters (based on biosynthetic knowledge) for further investigation. Cheng et al extensively profile the products of the selected biosynthetic gene clusters (BGCs) using a set of diverse methods, including spectrometric, spectroscopic, and molecular biology approaches. The prioritized BGC products also show promising antibiotic activity in both in vitro and in vivo models and could have potential for further antibiotic drug development.

The study by Cheng et al is highly interesting, original and relevant to both a specialist and more general audience. The antibiotic resistance crisis is of societal concern, and the newly reported molecules appear to be surprisingly effective against multiple drug-resistant Gram-positive strains. The authors provide a concise narrative with a clear hypothesis, targeting a specific class of molecules and obtaining promising results. Particularly satisfying is the diligence the authors applied in providing experimental evidence for connecting BGC to metabolites and defining their chemical structures. The literature was surveyed thoroughly, and relevant papers are referenced appropriately. Several minor errors and inaccuracies were detected, which are addressed in the Comments below. Overall, I recommend this manuscript to be accepted with minor revisions by Nature Communications.

General Comments:

- Cheng et al describe several interesting BGCs (mat, sis, keb) and characterize them appropriately. However, the exact loci of the BGCs in the bacterial genomes are described insufficiently: for example, in case of the mat gene cluster from *Streptomyces leeuwenhoekii* DSM42122, it is unclear in which genome with which GenBank reference this BGC can be found. Interested readers would need to dig through the supplemental information to find the particular precursor peptide, and BLAST the precursor-encoding gene against the *Streptomyces leeuwenhoekii* genome to find the locus and then search manually in its vicinity to identify the BGC. This is inconvenient and I would recommend to briefly summarize key information about the BGC (i.e. at least GenBank reference of the genome, start and stop coordinates of the locus) in a single place to make the BGC more easily searchable. Even better would be the submitting this information to the MIBiG (Minimum Information about a Biosynthetic Gene Cluster - <https://mibig.secondarymetabolites.org/>) database and referencing its MIBiG BGC accession number in the manuscript text. MIBiG has become the de-facto standard database for characterized BGCs, is freely accessible and open source, and adheres to FAIR principles (Findable, Accessible, Interoperable, Reusable). For example, the cypemycin BGC, which is very similar to the newly reported BGCs, can be

also found on MIBiG (see <https://mibig.secondarymetabolites.org/repository/BGC0000582/index.html#r1c1>). MIBiG relies on user contributions to grow, and making the newly described BGCs part of MIBiG would further increase the scientific impact of the manuscript.

Specific Comments:

- p(age) 3 introduction: in lines 8 and 9, several RiPP and NRP metabolites are described. However, two of the arguably most important anti Gram-positive antibiotics, nisin (RiPP) and vancomycin (NRP) are not mentioned in this place, even though they are highly relevant and also referenced later in the results (see antibiotic activity testing). I think it would be appropriate to reference them already in the introduction.
- In the introduction, the authors briefly mention that the genomes used for mining are publicly available. In the results, they describe that they used 21,911 genomes of actinobacteria and 30,666 genomes of firmicutes for the analysis. However, it is not clear which genomes these are and from where they were sourced. This information would be crucial for anybody who would want to replicate the described analysis workflow. Looking at Supplementary Dataset 1, many of the flavoproteins seem to have RefSeq IDs, while some have IDs that seemed to have been modified in some form (e.g. gbKHF30819.1 seems to be a GenBank identifier with a prefixed gb). It would be interesting to have a brief explanation on how and why the genomes under investigation were selected (e.g. RefSeq genomes for Actinobacteria and Firmicutes) and which genomes these were (e.g. a list of GenBank identifiers of the used genomes). Providing this information would help to make the study reproducible.
- On page 7 and 8, the authors investigate the mat BGC. The first paragraph describes the mat BGC in detail, followed by a paragraph of pinpointing the expected metabolite m/z in LC-MS(/MS) data. This sudden change is confusing – how were the samples generated on which LC-MS analysis was performed? Reading the methods, it becomes clear that the strain was first cultivated conventionally, followed by extraction of the mycelium and the growth medium, and followed by LC-MS analysis. Here, a sentence on the cultivation would help to retain the narrative flow of BGC → extract → LC-MS(/MS) analysis.
- In Figure 2a, the authors display a sequence similarity network subcluster of 4 nodes in which one node is annotated as microbisporicin and one node is annotated as NAI-107. However, these are synonyms of the same molecule (see Castiglione 2008, Foulston 2010 for a comparison of the molecules). Looking at Supplementary Dataset 1, precursor peptide sequences of peptides designated as microbisporicin and NAI-107 are indeed identical, albeit the sequences of the flavoproteins are not. It appears that the microbisporicin/NAI-107 precursor peptide can be found in multiple genomes. Nonetheless, the annotation of the sequence similarity network must be changed from “microbisporicin” and “NAI-107” to “microbisporicin/NAI-107” to make it clear that these are synonyms and not distinct molecules.
- In Figure 2c, the labels of the “2-Cys-containing precursors” “3-Cys-containing precursors” are formulated misguidedly since the “2” and “3” resemble chemical notations. It should be formulated as “precursors with 2 Cys” and “precursors with 3 Cys”.
- On page 10, line 6: typo in “modification” (“modifcation”).
- On page 11, line 1: the second part of the sentence is unclear and may be clarified.
- On page 15, line 26: suggested to add “possible” (“suggesting a possible synergistic effect”)
- On page 23, last line: incorrect syntax - “survival rate was recorded.” instead of “and recorded the

survival rate”.

- References: The authors did not reference the paper by Arnison et al 2013 (DOI: 10.1039/C2NP20085F). Given the importance of this paper on RiPP nomenclature and classification, I would suggest adding this reference for completeness.

Reviewer #2 (Remarks to the Author):

The manuscript describes a genome mining approach looking at co-occurrence of two genes to explore the chemical space of AviCys-containing RiPP peptides. While a small number of Avi(Me)Cys-containing RiPPs have been identified previously, the authors considerably expand this family by identifying over one thousand such precursor peptides in the genomic databases. After genome mining, heterologous expression and targeted metabolomic analysis were used to (partially) characterize three Avi(Me)Cys- and three AviCys-containing peptides. Interestingly, one of these peptides displayed strong antibacterial activity against Gram positive bacteria.

The genome-mining approach used by the authors is clever and it can inspire others to devise similar strategies. However, many authors have previously calculated all possible m/z values expected from the different posttranslational modifications to navigate the metabolome of the producing strain, so this approach is not novel.

The main issue with the manuscript is that its novelty should be clearly presented, as it is unclear whether it is the discovery workflow, unprecedented structures of natural products or their antibacterial activity. In this reviewer's opinion, the story about massatide A is almost complete: the authors report its identification, heterologous expression, full NMR data, and extensive characterization of its antibacterial properties. The characterization of two members of the most abundant "sis" family is however incomplete: NMR assignments are incomplete, the promiscuity of the maturase is superficially explored, and the antibacterial properties give incomplete, and sometime contradictory results. Is this part of the manuscript really necessary?

There are additional flaws in the manuscript, with experimental details missing, misleading statements and inappropriate reporting of literature. Some examples are reported below:

1. Abstract: the authors refer to "massatide A" without explaining that it is a RiPP described in this work. Also, there is no mention of the RiPP families identified, which represent a major finding by the authors.
2. Introduction, first paragraph: not clear what the authors mean by "conventional small-molecule antibiotics" as opposed to ribosomally and NRPS-made peptides: penicillin, cephalosporin, bacitracin and vancomycin are all clinically approved, NRPS-synthesized peptides!
3. Introduction, second paragraph: The shortage of new antibiotics is mostly due to a dysfunctional market, and not to the inability to discover new ones by the "grind and find approach".
4. The acronym ACP is used in the field to denote the acyl carrier protein (domain) involved in polyketide biosynthesis, so I would select a different one. In addition, the presence of Aminovinyl-(methyl)-cysteine implies the existence of a cycle, so referring to "cyclo" in the name is not really necessary.
5. There are several aminovinyl-(methyl)-cysteine-containing peptides described. The authors seem to cherry pick which ones to show in Figs 1 and S1, without consistency between the figures and with the

text. More examples appear in Fig.2.

6. p.4, first paragraph: the example of microbisporicins is misleading, as it seems that it is the aminovinyl-(methyl)-cysteine-containing ring that results in its mechanism of action, while it has been shown that it is the lack of a negative charge that enhances potency.
7. Fig. 2a: it is not clear how the hits identified by genome mining were clustered into families. Did the authors consider the entire BGC or just the predicted core peptide? In addition, the authors rightly include known peptides but do not mention Prestide A1 from the sis family. Also, microbisporicin and NAI-107 are the same molecule. Given the large number of aminovinyl-(methyl)-cysteine-containing peptides identified by their genomic mining approach, this section should be expanded by expanding on the bioinformatically-predicted chemical diversity, as compared with current knowledge.
8. Pages 8 and 10, it was unclear whether full gene cluster or which genes were expressed during heterologous expression described in the main text.
9. Page 15, the authors conclude that the maturase is tolerant of core peptide structures. But based on the Fig. S34-35, the modified peptides are present in very low yields. Page 21, the statement has oversold the presented data in Fig. S34-35 "Interestingly, the sis family of ACPs serves as a noteworthy example, illustrating the remarkable substrate tolerance within ACP biosynthesis. This observation offers a promising avenue for bioengineering new-to-nature analogs with desired properties and creating novel antibiotics".
10. The MICs of nisin reported in Table 1 are very high and do not correspond to values reported in the literature! In addition, the authors should discuss/compare the properties of massatide A with those reported of lanthipeptides brought in advanced preclinical development (e.g., gallidermin, NAI-107, NVB302). Also, there is no discussion about the possible mechanism of antibacterial activity.
11. p.15, last line: the term "efficacy" is used for animal studies, not for in vitro studies
12. p. 16: the data that sistertide A1 and A2 act synergistically are not convincing - see also main comment above.
13. p.17, middle paragraph: "immense potential" is not appropriate
14. Discussion, first paragraph: "We successfully discovered five ACPs that exhibited potent antibacterial activity". Actually, only massatides do.
15. The sentence "Among the uncharacterized BGCs in our dataset, we identified certain BGCs containing a protein with a DUF3105 domain (Figure S39).", if important, should be part of Results and properly explained, not thrown in in the Discussion.
16. In the Discussion, the authors refer to water solubility issues for massatide A, but have not shown any data.
17. What was the concentration of massatide A before administration to animals?
18. How did the authors establish chirality from the data reported in Tables S8–S11.

Reviewer #3 (Remarks to the Author):

The authors describe the heterologous expression and purification of five macrocyclic peptides. Thereof, two compounds, the massatides A and B, showed promising MIC values against Gram-positive bacteria. Noteworthy, massatide A was tested in vivo using a mouse septicemia model, in which the infection was caused by MRSA.

The authors screened genome data to select putative biosynthetic gene clusters (BGCs) related to known

Aminovinyl-(methyl)-cysteine-containing cyclopeptides, which are RiPPs. The in silico screen of available whole genome data showed that only a low percentage of precursor molecules is to date experimentally described. They selected three different BGCs and analyzed the wild type strains for production of the predicted metabolites. In two out of the three cases, the metabolites could be directly dereplicated from the crude extracts. Furthermore, they successfully heterologously expressed the three BGCs in a *Streptomyces albus* host. This enabled purification of metabolites corresponding to each of the three BGCs.

Overall, the manuscript describes nicely a natural product discovery workflow for the detection of potential lead structures for future antibiotic development. By reading the manuscript, this reviewer had the feeling that a few points could be considered to make the statements more clear, or to adjust the wording a bit.

In the introduction section, it is stated that the scientific community has turned its attention to antimicrobial peptides (AMPs) as a potential solution for new antibiotics in addition to small molecules. The definition of AMPs is not clear and causes difficulties. My feeling would be that usually AMPs are regarded as ribosomally synthesized peptides with a certain length (e.g., depending on the definition colistin might be a member or not....mostly people would even not directly think about thiostrepton). Therefore, the smaller molecules, e.g. darobactin, dynobactin, as well as the nonribosomal peptides mentioned are not fitting. It could be better to directly refer to the term “macrocylic peptides” used in the next sentence.

Then, it is stated that the grind and find approach is inadequate. This is a strong word, since the antibiotics in use were discovered that way, and the work presented by the authors could be also seen as the same approach. Some compounds were produced and then tested for their bioactivity. Of course, I totally agree that a rationale-based prioritization is needed to enhance probability of success. The abbreviation “ACPs” could be reconsidered (ACyPs?), since with the NRPs also modular systems are mentioned and there ACPs could be acyl carrier proteins.

The advantage of the mass mapping pipeline is not directly clear. It is stated that this helped to swiftly identify the relevant BGC before isolation. However, the authors used heterologous expression to isolate the metabolites of interest. Then, the BGC is already clear. Out of curiosity, were there many metabolites with such a high molecular weight present in the extracts and so many putative BGCs present in the genomes that it is unclear if the metabolite can be linked to a corresponding BGC by MS/MS analysis? Furthermore, for the keb cluster the analysis of the wild type strain did not show any predicted metabolite. Then, it was directly used for heterologous expression. Reasonable to use heterologous approach, since it was working for the others. However, it somehow could break the logic of the workflow described in the manuscript.

In the results section:

The sentence “Given the presence of these two precursor subfamilies within one BGC” is somehow unclear. Are there always 2 precursors, one with 2 and one with 3 Cys residues? I agree with the point that “that a single maturation system within the sis family could generate ACPs with diverse thioether rings”; however, this is based on the specific precursors within one BGC (maybe the word families makes it hard to read as the sentence is structured right now).

Is there any specific reason why only MatR1 and not MatR2 was analyzed experimentally?

In the figures the crude extracts of the producer strains are not shown, only of the heterologous producer strains. Is this due to the fact that the expression level is too low in the wild type strains? Maybe it could be mentioned how concentrated the tested extracts were (since the inhibition zones are quite small).

“one tailoring enzyme system constructs multiple crosslinking patterns”. Can this be stated in this way? Is the enzyme identified in the different BGCs identical, or homologous?

Antibacterial activity. It is stated that “slightly larger inhibition zone in Figure 4e, suggesting a synergistic effect”. Could be somehow contradictory. Is it clear that it is a synergistic and not an additive effect? (The same for kebanetides A1 and A2)

This point should be clarified throughout the manuscript. A checkerboard assay using the purified compounds could be done.

“potential as a promising candidate for gram-positive antibiotics.” Might be a far fetch at this stage of the project. Maybe it can be toned down a bit.

Efficacy in vitro.

I understand the point that only an 8-fold increase on the MIC was reached within 6 days. However, absolutely no shift was seen for the control vancomycin, even so resistance is reported and should occur in *S. aureus*. (Some numbers in an old article from 2002 “An Elevated Mutation Frequency Favors Development of Vancomycin Resistance in *Staphylococcus aureus*”). Maybe some more words as explanation could help here. Anything known about the mode of action? (I assume it will be related to other members of the Aminovinyl-(methyl-)cysteine-containing cyclopeptides)

“massatide A showed low toxicity to Hela cells only at a high concentration of 200 $\mu\text{g}/\text{mL}$ ”. Is this wording correct? From 40 to 200 $\mu\text{g}/\text{mL}$ the viability of Hela cells dropped from 1.0%? to around 0.25 (Figure S38). Can this be regarded as low? Furthermore, the cytotox against Hek293T cells looks stronger, even at 8 $\mu\text{g}/\text{mL}$ a clear decrease in viability. This would indicate a rather small therapeutic window. In Figure S38 E it seems like the values were exchanged between massatide A and B. Furthermore, is the IC50 against Hela cells indeed >200 (see point before)?

Figure 5. Here it would be nice to get some later time points. In a) it can be seen that the curve for the 1xMIC starts to rise as well. Even more important, why are the data missing in c)?

It would be highly interesting to see how the CFUs develop. Will there be a rebound? Will there be resistance development visible in these experiments?

In the discussion, the point “with low toxicity and minimal risk of resistance” is repeated. However, it is questionable if low tox is really the appropriate term.

In summary, the in vivo data are convincing. However, it would be good to know which compound levels were reached within the mouse. Maybe this could be used for the argumentation about the cytotox issues observed before (keeping the observed cytotox against liver cells in mind).

Reviewer #4 (Remarks to the Author):

“I co-reviewed this manuscript with one of the reviewers who provided the listed reports. This is part of the Nature Communications initiative to facilitate training in peer review and to provide appropriate recognition for Early Career Researchers who co-review manuscripts.”

Point-by-point response letter.

Reviewer Comments:

Reviewer #1

In this study, Cheng et al describe the discovery of several new ribosomally synthesized and post-translationally modified peptides (RiPPs) with antibiotic properties. The authors apply their recently described SPECO approach (Short Peptide and Enzyme Co-Occurrence analysis pipeline) to mine publicly available genome data for short peptide-encoding genes that co-occur with a specific flavoprotein. This enzyme is assumed to introduce a aminovinyl-(methyl-)cysteine moiety, which is a modification found in several families of RiPPs, most prominently in the linaridin class of antibiotics. The authors use their bioinformatics pipeline to detect small putative precursor peptides co-occurring in the vicinity of such flavoproteins, cluster them based on sequence similarity of the precursor peptides, and prioritize candidates from two promising clusters (based on biosynthetic knowledge) for further investigation. Cheng et al extensively profile the products of the selected biosynthetic gene clusters (BGCs) using a set of diverse methods, including spectrometric, spectroscopic, and molecular biology approaches. The prioritized BGC products also show promising antibiotic activity in both in vitro and in vivo models and could have potential for further antibiotic drug development.

The study by Cheng et al is highly interesting, original and relevant to both a specialist and more general audience. The antibiotic resistance crisis is of societal concern, and the newly reported molecules appear to be surprisingly effective against multiple drug-resistant Gram-positive strains. The authors provide a concise narrative with a clear hypothesis, targeting a specific class of molecules and obtaining promising results. Particularly satisfying is the diligence the authors applied in providing experimental evidence for connecting BGC to metabolites and defining their chemical structures. The literature was surveyed thoroughly, and relevant papers are referenced appropriately. Several minor errors and inaccuracies were detected, which are addressed in the Comments below. Overall, I recommend this manuscript to be accepted with minor revisions by Nature Communications.

Response: We are grateful for the positive feedback and valuable suggestions provided by this reviewer. We have carefully considered each point and made revisions accordingly. Below, we present our point-by-point responses, aiming to address the majority, if not all, of the comments raised.

1. Cheng et al describe several interesting BGCs (mat, sis, keb) and characterize them appropriately. However, the exact loci of the BGCs in the bacterial genomes are described insufficiently: for example, in case of the mat gene cluster from *Streptomyces leeuwenhoekii* DSM42122, it is unclear in which genome with which GenBank reference this BGC can be found. Interested readers would need to dig through the supplemental information to find the particular precursor peptide, and

BLAST the precursor-encoding gene against the *Streptomyces leeuwenhoekii* genome to find the locus and then search manually in its vicinity to identify the BGC. This is inconvenient and I would recommend to briefly summarize key information about the BGC (i.e. at least GenBank reference of the genome, start and stop coordinates of the locus) in a single place to make the BGC more easily searchable.

Response 1: To address this concern, we have augmented Table S3 with the following information: strain name, assembly accession (for retrieving genome from NCBI), loci of BGC, start and end coordinates of BGC, orf annotations, including protein accession ID, protein sequence, length, conserved domain, and proposed function. We believe that this additional information will assist interested readers in identifying these BGCs effectively. Below is an example of the updated structure of Table S3:

mat BGC					
strain	assembly	locus	start	end	
Streptomyces leeuwenhoekii DSM42122	GCF_001013905.1	NZ_LN83 1790.1	7775129	7788598	
orf annotation					
orf	Accession ID	Sequence	Length	Conserved domain	Proposed function
matA	WP_164497196.1	MDTHELIEGFDAYVEAELNEDAMVDAPATTVPCTV ASFATGYFSC*	46	LxA domain RiPP leader family	Precursor peptide
matD	WP_047122252.1	MTAAAAGEKAPGAPRETRTLVVVTGSLSAAFVPPQGL GHLRMTHPGVRIITLITRSALKFVTATAVAAATGGEVT LDAWEDDRPGTPAEALHVELAEWAERIVVYPASWHY VARLAQGLADAPSLALHTSRAEVAVAPSVPPGSLDS KVFQRHLAELDRGYRVAPVLLARSTGGTRPALVPPP MPDVMALFDQAPPAPSPAPAGRQAAP*	211	Flavoprotein, PF02441	Cysteine decarboxylase

- Even better would be the submitting this information to the MIBiG (Minimum Information about a Biosynthetic Gene Cluster - <https://mibig.secondarymetabolites.org/>) database and referencing its MIBiG BGC accession number in the manuscript text. MIBiG has become the de-facto standard database for characterized BGCs, is freely accessible and open source, and adheres to FAIR principles (Findable, Accessible, Interoperable, Reusable). For example, the cypemycin BGC, which is very similar to the newly reported BGCs, can be also found on MIBiG (see <https://mibig.secondarymetabolites.org/repository/BGC0000582/index.html#r1c1>). MIBiG relies on user contributions to grow, and making the newly described BGCs part of MIBiG would further increase the scientific impact of the manuscript.

Response 2: We completely agree with this suggestion. We have submitted the *mat*, *sis* and *keb* BGCs to the MIBiG database. Each BGC has been temporarily assigned an accession number as follows:

BGC	Accession ID
mat	BGC0002835
keb	BGC0002836
sis	BGC0002837

However, during the preparation of this response letter, the submission system of MIBiG for accessing full details was temporarily unavailable. Consequently, the detailed BGC information had to be manually formatted by Prof. Marnix H. Medema's team. As a result, access to these BGCs via MIBiG accession ID may experience delays.

3. p(age) 3 introduction: in lines 8 and 9, several RiPP and NRP metabolites are described. However, two of the arguably most important anti Gram-positive antibiotics, nisin (RiPP) and vancomycin (NRP) are not mentioned in this place, even though they are highly relevant and also referenced later in the results (see antibiotic activity testing). I think it would be appropriate to reference them already in the introduction.

Response 3: Thank you for the suggestion. We have added nisin and vancomycin in the introduction and cited references accordingly (lines 46-47).

4. In the introduction, the authors briefly mention that the genomes used for mining are publicly available. In the results, they describe that they used 21,911 genomes of actinobacteria and 30,666 genomes of firmicutes for the analysis. However, it is not clear which genomes these are and from where they were sourced. This information would be crucial for anybody who would want to replicate the described analysis workflow. Looking and Supplementary Dataset 1, many of the flavoproteins seem to have RefSeq IDs, while some have IDs that seemed to have been modified in some form (e.g. gbKHF30819.1 seems to be a GenBank identifier with a prefixed gb). It would be interesting to have a brief explanation on how and why the genomes under investigation were selected (e.g. RefSeq genomes for Actinobacteria and Firmicutes) and which genomes these were (e.g. a list of GenBank identifiers of the used genomes). Providing this information would help to make the study reproducible.

Response 4: Thank you for bringing up these points. We have made the following revisions to clarify the dataset we used.

“However, it is not clear which genomes these are and from where they were sourced.”

These genomes were downloaded from the NCBI database (Data: 2021-08-11). We provided additional genome information in supplementary dataset 1 under the sheet “**21911_actinobacteria**” and “**30666_firmicutes**,” which contain the **strain name** and **assembly number** (which can be used to retrieve genome from NCBI) of 21,911 genomes of actinobacteria and 30,666 genomes of firmicutes. Below is the data structure of this file:

1	21911 actinobacteria genomes with strain names and assembly numbers.	
2	Assembly number can be used to retrieve genome from NCBI.	
3	strain name	assembly number
4	Acaricomes_phytoseiuli_DSM_14247	GCF_000376245.1
5	Acidimicrobium_ferrooxidans_DSM_10331	GCF_000023265.1
6	Acidipropionibacterium_acidipropionici_ATCC_4875	GCF_000310065.1
7	Acidipropionibacterium_acidipropionici_ATCC_55737	GCF_001602115.1
8	Acidipropionibacterium_acidipropionici_CGMCC_1.2230	GCF_001441165.1
9	Acidipropionibacterium_acidipropionici_DSM_4900	GCF_000427845.1

We have included additional information regarding the source of genomes in the main text (lines 83-85):

“.....we conducted a large-scale analysis of publicly available actinobacteria and firmicutes genomes using the rule-based genome mining pipeline SPECO²¹ (Figure 1b and supplementary dataset 1).”

“Looking and Supplementary Dataset 1, many of the flavoproteins seem to have RefSeq IDs, while some have IDs that seemed to have been modified in some form (e.g. gbKHF30819.1 seems to be a GenBank identifier with a prefixed gb).”

In the raw data, we previously deleted “ref” and “|” in the 3rd column to make the accession more readable. This process led to the presence of ‘gbxxxxxxx’ and the absence of ‘ref’.

Now, in the revised Supplementary Dataset 1, we have provided the original raw data and gave a “Note” to the format of protein accession ids. You may refer to the revised data below:

1	Note: The 'accession id' column contains the protein IDs either from RefSeq or GeneBank. The 'ref ' and 'gb ' in the id respectively denotes the source of the IDs.			
2				
3				
4		flavo_seq	accession id	pre_seq
5	firmicutes_pre_1	MRFAGKTVVLCVTGGIA	ref WP_117555070.1	MFGFIRPVKAELRVKE/ firmicutes
6	firmicutes_pre_2	MRFAGKTVVLCVTGGIA	ref WP_117488806.1	MFGFIRPVKAELRVKE/ firmicutes
7	firmicutes_pre_3	MIPQVADGLRGKKITIGV	ref WP_115792525.1	MEKNWRMYASLDAD firmicutes
8	firmicutes_pre_4	MELKELVEQIAVKVAERI	ref WP_128746479.1	MIFKHESARIYHEDQN firmicutes
9	firmicutes_pre_5	MNRKEFEDALMATIVEY	ref WP_005344390.1	MYLGKVGITVVSTIKTP firmicutes
10	firmicutes_pre_9	MLEGKNILLGVSGGIAAY	ref WP_015777891.1	MNNPSFKKLEISPSRY firmicutes
11	firmicutes_pre_11	MLEGKNILLGVSGGIAAY	ref WP_105300500.1	MNNPSFKKLEISPSRY firmicutes
12	firmicutes_pre_14	MNRKEFEELMSTIVEY	ref WP_096241298.1	MYLGKVGITVVSTVKA firmicutes
13	firmicutes_pre_15	MLEEKNILLGITGGIASYK	ref WP_060929560.1	MNNPSFKKLEISPSRY firmicutes
14	firmicutes_pre_16	MNRKEFEDALMATIVEY	ref WP_118313830.1	MYLGKVGITVVSTIKTP firmicutes

“It would be interesting to have a brief explanation on how and why the genomes under investigation were selected (e.g. RefSeq genomes for Actinobacteria and Firmicutes) and which genomes these were (e.g. a list of GenBank identifiers of the used genomes).”

In this manuscript, we focused solely on the analysis of AviCys RiPP BGCs from actinobacteria and firmicutes for two primary reasons. Initially, we aimed to analyze bacterial genomes across various phyla, including actinobacteria, firmicutes, proteobacteria, and others. However, our preliminary investigation revealed that AviCys RiPP BGCs were predominantly present in the phyla of actinobacteria and

firmicutes. Additionally, we noted that previously reported AviCys-containing RiPPs were primarily derived from these two phyla. In response to the reviewer's suggestion, we have added a brief explanation, “We noticed that ACyPs were mainly found from firmicutes and actinobacteria, the genomes from these two phyla were then selected and analyzed.” to the first paragraph of the results section in lines 107-108.

5. On page 7 and 8, the authors investigate the mat BGC. The first paragraph describes the mat BGC in detail, followed by a paragraph of pinpointing the expected metabolite m/z in LC-MS(/MS) data. This sudden change is confusing – how were the samples generated on which LC-MS analysis was performed? Reading the methods, it becomes clear that the strain was first cultivated conventionally, followed by extraction of the mycelium and the growth medium, and followed by LC-MS analysis. Here, a sentence on the cultivation would help to retain the narrative flow of BGC → extract → LC-MS(/MS) analysis.

Response 5: We appreciate the suggestion. In line with this recommendation, we have inserted a sentence at the beginning of the second paragraph in lines 146-148. “The wild type strain *Streptomyces leeuwenhoekii* DSM42122 was cultivated for 7 days and the high-resolution mass spectrometry (HRMS) data of the crude extract was collected.”

6. In Figure 2a, the authors display a sequence similarity network subcluster of 4 nodes in which one node is annotated as microbisporicin and one node is annotated as NAI-107. However, these are synonyms of the same molecule (see Castiglione 2008, Foulston 2010 for a comparison of the molecules). Looking at Supplementary Dataset 1, precursor peptide sequences of peptides designated as microbisporicin and NAI-107 are indeed identical, albeit the sequences of the flavoproteins are not. It appears that the microbisporicin/NAI-107 precursor peptide can be found in multiple genomes. Nonetheless, the annotation of the sequence similarity network must be changed from “microbisporicin” and “NAI-107” to “microbisporicin/NAI-107” to make it clear that these are synonyms and not distinct molecules.

Response 6: Thank you for bringing this to our attention. We have taken the following steps to address this issue:

- (1) We have revised the original SSN, as shown below.
- (2) To clarify this point, We have added the clarification "microbisporicin and NAI-107 are synonyms and not distinct molecules" in the Figure 2 legend. Additionally, we have corrected "Pristide A1" in the sis cluster to "Pristinin A3" based on the reference (Alexander M. Kloosterman et al., doi.org/10.1371/journal.pbio.3001026).

Please find below the comparison between the original and revised Figure 2a:

The original SSN in Figure 2a:

The revised SSN in Figure 2a:

7. In Figure 2c, the labels of the “2-Cys-containing precursors” “3-Cys-containing precursors” are formulated misguidedly since the “2” and “3” resemble chemical notations. It should be formulated as “precursors with 2 Cys” and “precursors with 3 Cys”.

Response 7: We have revised the labels to “precursors with 2 Cys” and “precursors with 3 Cys” throughout the manuscript.

The original Figure 2c:

The revised Figure 2c:

8. On page 10, line 6: typo in “modification” (“modification”).

Response 8: We have revised “modification” to “modification” in line 185.

9. On page 11, line 1: the second part of the sentence is unclear and may be clarified.

Response 9: We have revised the original statement “We next investigated the *sis* family, which is predicted to catalyze two distinct crosslinking patterns, aiming to convert its biosynthetic potential into peptide antibiotics.” to “We next investigated the *sis* family, which is anticipated to produce ACyPs containing 2 and 3 thioether rings. This is due to the existence of precursor peptides containing 2 and 3 cysteine residues.” in lines 206-207.

10. On page 15, line 26: suggested to add “possible” (“suggesting a possible synergistic effect”)

Response 10: In light of the suggestions from the other two reviewers, we agree that the current data cannot support the existence of a “synergistic effect” in this manuscript. Therefore, we have chosen to remove the conclusion regarding any synergistic effect.

11. On page 23, last line: incorrect syntax - “survival rate was recorded.” instead of “and recorded the survival rate”.

Response 11: We revised this sentence to “survival rate was recorded” in line 476.

12. References: The authors did not reference the paper by Arnison et al 2013 (DOI: 10.1039/C2NP20085F). Given the importance of this paper on RiPP nomenclature and classification, I would suggest adding this reference for completeness.

Response 12: We added this reference in the introduction, line 56, lines 545-546 “.....offering a remarkable diversity of structures and bioactive potential for

antibiotic discovery^{17,18}”.

Reviewer #2

The manuscript describes a genome mining approach looking at co-occurrence of two genes to explore the chemical space of AviCys-containing RiPP peptides. While a small number of Avi(Me)Cys-containing RiPPs have been identified previously, the authors considerably expand this family by identifying over one thousand such precursor peptides in the genomic databases. After genome mining, heterologous expression and targeted metabolomic analysis were used to (partially) characterize three Avi(Me)Cys- and three AviCys-containing peptides. Interestingly, one of these peptides displayed strong antibacterial activity against Gram positive bacteria.

The genome-mining approach used by the authors is clever and it can inspire others to devise similar strategies. However, many authors have previously calculated all possible m/z values expected from the different posttranslational modifications to navigate the metabolome of the producing strain, so this approach is not novel.

Response: We are grateful for both the positive comments and informative suggestions provided by this reviewer. Below, please find our point-by-point responses, which we believe address all the comments raised.

The main issue with the manuscript is that its novelty should be clearly presented, as it is unclear whether it is the discovery workflow, unprecedented structures of natural products or their antibacterial activity. In this reviewer's opinion, the story about massatide A is almost complete: the authors report its identification, heterologous expression, full NMR data, and extensive characterization of its antibacterial properties. The characterization of two members of the most abundant "sis" family is however incomplete: NMR assignments are incomplete, the promiscuity of the maturase is superficially explored, and the antibacterial properties give incomplete, and sometime contradictory results. Is this part of the manuscript really necessary?

Response: We appreciate the reviewer's feedback regarding the clarity of novelty in our manuscript. To address this, we would like to emphasize that while biosynthetic logic-guided strategies and mass spectrometry-guided approaches have been individually employed for genome mining of biosynthetic gene clusters, our manuscript introduces a practical approach by combining both strategies. By merging these methodologies, we successfully tackled a large dataset, prioritizing Avi(Me)Cys-containing RiPP antibiotics from a pool of over 50,000 bacterial genomes. This synergistic strategy proved highly effective, leading to the identification of a structurally novel and antibacterially effective antibiotic named massatide A.

In addition, we included the identification of the *sis* family of Avi(Me)Cys-containing antibiotics in this manuscript, despite incomplete NMR characterization, to illustrate the potential of this strategy. The low yield of compounds **3**, **4**, and **5** posed

challenges to their full characterization and *in vivo* assay, despite our best efforts to infer their structures from available data (refer to the table below).

	Ribosomal origin	HRMS	Full NMR	Advanced Marfey
Compound 2	✓	✓	✓	✓
Compound 3	✓	✓		
Compound 4	✓	✓	Proton NMR only	✓
Compound 5	✓	✓	Proton NMR only	✓

We acknowledge that the characterization of the *sis* family is incomplete due to their low production yields, which hindered full NMR analysis. Thus, we have revised “structures” to “suggested structures” in figure 4 (line 243) to tone down the characterization statement of the *sis* family compounds. Nevertheless, we deemed it necessary to include this section in our manuscript for several reasons. Firstly, it is crucial to demonstrate the practicality and value of our method and address any concerns about cherry-picking from our readers. Secondly, while this platform was utilized for one case, it has the potential to be applied to numerous other RiPPs. Additionally, characterizing the two *sis* family BGCs is important for understanding the biosynthesis, bioengineering, and potential biological activities of these compounds. The leader-core-enzyme mix-and-match assay is a straightforward experimental design that has revealed the bioengineering potential of these maturases for diversifying ACyPs. We hope that our preliminary research will benefit not only researchers in the antibiotic discovery field but also scientists in the bioengineering domain. However, we acknowledge that further extensive investigation of the maturases' promiscuity is necessary to explore the bioengineering potential of this family in the future.

There are additional flaws in the manuscript, with experimental details missing, misleading statements and inappropriate reporting of literature. Some examples are reported below:

1. Abstract: the authors refer to "massatide A" without explaining that it is a RiPP described in this work. Also, there is no mention of the RiPP families identified, which represent a major finding by the authors.

Response 1: We revised the original sentence in the abstract “Notably, massatide A displayed excellent activity against a spectrum of gram-positive pathogens.....” to “Notably, we identified a class V lanthipeptide, massatide A, which displayed excellent activity against a spectrum of gram-positive pathogens.....” in lines 22-23.

2. Introduction, first paragraph: not clear what the authors mean by "conventional small-molecule antibiotics" as opposed to ribosomally and NRPS-made peptides: penicillin, cephalosporin, bacitracin and vancomycin are all clinically approved, NRPS-synthesized peptides!

Response 2: We apologize for any confusion caused by our original statement. Our intention was not to suggest that "small-molecule antibiotics" are the opposite of "ribosomally and NRPS-made peptides." Rather, we wanted to highlight that

macrocyclic peptides offer an alternative source to antibiotic discovery. To clarify this, we have revised the original statement “In addition to conventional small-molecule antibiotics, the scientific community has turned its attention to antimicrobial peptides as a potential solution.” to “In addition to small-molecule antibiotics, macrocyclic peptides serve as a complementary alternative for the lead discovery.” in lines 41-42.

3. Introduction, second paragraph: The shortage of new antibiotics is mostly due to a dysfunctional market, and not to the inability to discover new ones by the "grind and find approach".

Response 3: We agree that the lack of incentives and investments for developing new antibiotics has contributed significantly to the shortage of novel antibiotics. Accordingly, we have revised the original statement “The traditional ‘grind and find’ approach, involving chemical and/or biological screenings to uncover natural product leads from field-collected resources or laboratory-cultured organisms in the quest for new antibiotics has proven to be both time-consuming and inadequate.” to “The traditional chemical and/or biological screening method in uncovering new antibiotics from field-collected resources or laboratory-cultured organisms has proven to be both time-consuming and inadequate.” in lines 50-51. In addition, we have removed the statement “This has resulted in a shortage of novel antibiotics to combat the ever-growing threat of antibiotic-resistant bacteria on a global scale.”

4. The acronym ACP is used in the field to denote the acyl carrier protein (domain) involved in polyketide biosynthesis, so I would select a different one. In addition, the presence of Aminovinyl-(methyl)-cysteine implies the existence of a cycle, so referring to "cyclo" in the name is not really necessary.

Response 4: Thank you for bringing this to our attention. Following the suggestion from reviewer 3, we have revised the acronym “ACP” to “ACyPs” (AviCys-containing peptides) and removed “cyclo” from the name.

5. There are several aminovinyl-(methyl)-cysteine-containing peptides described. The authors seem to cherry pick which ones to show in Figs 1 and S1, without consistency between the figures and with the text. More examples appear in Fig.2.

Response 5: In response to your comment, we have updated Figures 1a and S1 to include the structures of microsporicin and lexapeptide, instead of epidermin. These structures represent the five families of ACyPs, including lanthipeptide, thioamitide, linaridin, lipolanthin, and lanthidin (also called class V lanthipeptide), as mentioned in the introduction of the main text.

6. p.4, first paragraph: the example of microbisporicins is misleading, as it seems that it is the aminovinyl-(methyl)-cysteine-containing ring that results in its mechanism of action, while it has been shown that it is the lack of a negative charge that enhances

potency.

Response 6: Thank you for bringing this to our attention. We have implemented your suggestion to revise these statements in lines 67-68. They now read as follows “This highly modified structure also confers drug-like properties to the compound, including the heat stability and high target specificity. Microbisporicin, which is an Avi(Me)Cys-containing lanthipeptide, interacts electrostatically with the negatively charged lipid II pyrophosphate bridge, making them effective against vancomycin-resistant bacteria”.

7. Fig. 2a: it is not clear how the hits identified by genome mining were clustered into families. Did the authors consider the entire BGC or just the predicted core peptide? In addition, the authors rightly include known peptides but do not mention Prestide A1 from the *sis* family. Also, microbisporicin and NAI-107 are the same molecule. Given the large number of aminovinyl-(methyl)-cysteine-containing peptides identified by their genomic mining approach, this section should be expanded by expanding on the bioinformatically-predicted chemical diversity, as compared with current knowledge.

Response 7: We appreciate the valuable comments received. We have carefully addressed each comment individually, as outlined below:

“Did the authors consider the entire BGC or just the predicted core peptide?”

The SSN analysis was performed based on precursor peptide sequence similarity including both the leader region and core region. We have clarified the statement in line 112 by revising it as follows “Our findings indicate that most putative precursors were not clustered with known precursors (Figure 2a) as shown in the precursor sequence similarity network (SSN)”.

“In addition, the authors rightly include known peptides but do not mention Prestide A1 from the *sis* family.”

We have clarified the statement in lines 121-122 by revising it as follows “We were also drawn to the largest cluster in the SSN, which contains pristin A3, and have designated this cluster as the *sis* BGC family.”

“Also, microbisporicin and NAI-107 are the same molecule.”

To address this, we have made three revisions accordingly:

- (1) We have revised the original SSN, as shown below.
- (2) To clarify this point, We have added the clarification "microbisporicin and NAI-107 are synonyms and not distinct molecules" in the Figure 2 legend.
- (3) Additionally, we have corrected "Pristide A1" in the *sis* cluster to "Pristinin A3" based on the reference (Alexander M. Kloosterman et al., doi.org/10.1371/journal.pbio.3001026).

Please find below the comparison between the original and revised Figure 2a:

The original SSN in Figure 2a:

Revised SSN in Figure 2a:

“this section should be expanded by expanding on the bioinformatically-predicted chemical diversity, as compared with current knowledge”

Thank you for the suggestion. To illustrate the chemical diversity of the identified BGCs, we focused on analyzing the four largest families. However, due to space limitations in the main text, we have included the data in the Supporting Information under Figure S2. Additionally, we have provided all the remaining clusters and sequence logos in Supplementary Dataset 2 and 3. Key information included: (1) Sequence logo of each precursor family, (2) Representative BGC architecture of each family and phylogenetic analysis using BiGSCAPE (under $-cutoff$ 0.8), (3) Protein accession ID of the flavoprotein and (4) Protein annotation of essential open reading frame (ORF) in BGC. Please refer to the Figures below:

- (A) SSN analysis of putative precursors and Cluster 1 (C1) to 14 (C14) were labeled
- (B) Chemical diversity of putative AviCys RiPPs from Cluster C1
- (C) Chemical diversity of putative AviCys RiPPs from Cluster C2
- (D) Chemical diversity of putative AviCys RiPPs from Cluster C3
- (E) Chemical diversity of putative AviCys RiPPs from Cluster C5

B

Precursor logos and BGC phylogenetic analysis, representative clade was shown below

Cluster C1, the *sis* family

Chemical diversity: BGCs from C1 cluster contain multiple precursors. Precursors with both 2 cysteine residues and 3 cysteine residues are found in these BGCs, implying the encoded products may contain 2 or 3 thioether bridges. Variations inside the AviCys and thioether rings largely expanded the chemical diversity.

C

Cluster C2

Chemical diversity: BGCs from C2 cluster contain one or two precursors. The putative rings were highlighted below. Variations inside the thioether ring 1 largely expanded the chemical diversity.

D

Cluster C3

No known case was reported from Cluster C3

Chemical diversity: BGCs from C3 represents a new subfamily of AviCys-containing RiPPs.

E

Cluster C5

No known case was reported from Cluster C5

Chemical diversity: BGCs from C5 represents a new subfamily of AviCys-containing RiPPs.

8. Pages 8 and 10, it was unclear whether full gene cluster or which genes were expressed during heterologous expression described in the main text.

Response 8: The full gene cluster was cloned into a vector and then heterologously expressed in *S. albus*. We have added Table S3 which provides comprehensive details of the entire BGCs. We have revised sentence to “we cloned and heterologously expressed the entire *mat* BGC in *Streptomyces albus* J1074 (Table S3)” (line 158),

“To determine the essentiality of both MatR1 and MatR2, we constructed a single gene knockout strain *S. albus*/pCZMAT_ARI” (line 199), and “we cloned and heterologously expressed the entire *sis* BGC in *S. albus* (Table S3)” (line 221).

9. Page 15, the authors conclude that the maturase is tolerant of core peptide structures. But based on the Fig. S34-35, the modified peptides are present in very low yields. Page 21, the statement has oversold the presented data in Fig. S34-35 “Interestingly, the *sis* family of ACPs serves as a noteworthy example, illustrating the remarkable substrate tolerance within ACP biosynthesis. This observation offers a promising avenue for bioengineering new-to-nature analogs with desired properties and creating novel antibiotics”.

Response 9: We agree that the modified peptides showed very low yields, and we recognize that our previous statement may have been overstated. To make the statement more appropriate, we have revised it to “Interestingly, the substrate-promiscuous maturases from the *sis* family offers a possibility for bioengineering new-to-nature analogs with potential antibiotic activity.” in lines 399-401.

10. The MICs of nisin reported in Table 1 are very high and do not correspond to values reported in the literature! In addition, the authors should discuss/compare the properties of massatide A with those reported of lanthipeptides brought in advanced preclinical development (e.g., gallidermin, NAI-107, NVB302). Also, there is no discussion about the possible mechanism of antibacterial activity.

Response 10: We also noticed that the MICs of nisin to *S. aureus* ATCC25923 and *B. subtilis* 168 are high. Previously, we used nisin purchased from Sigma (CAS: 1414-45-5 Cat. No.: N5764). We proposed that the different MIC may be because of two reasons: Firstly, the MIC value may vary in different strains of indicator bacteria within the same species. Secondly, the purity of the nisin we used may have been insufficient. The HRMS spectrum below indicates the presence of nisin at 9.1 min, but with some impurities in the sample. While revising this manuscript, we also tested another nisin purchased from APExBIO (CAS: 1414-45-5 Cat. No.: C3531), but the MIC values remained high. The MICs ($\mu\text{g/mL}$) are shown below:

	Nisin (from Sigma)	Nisin (from APExBIO)
S. aureus ATCC25923	64	>64
M. luteus DSM1790	0.5	4
B. subtilis 168	32	64
E. faecalis OG1RF	>64	>64
E. faecium MCC2763	>64	>64

The HRMS result of the Nisin from sigma is shown below:

“In addition, the authors should discuss/compare the properties of massatide A with those reported of lanthipeptides brought in advanced preclinical development (e.g., gallidermin, NAI-107, NVB302).”

Following this suggestion, we have included Gallidermin, NAI-107 (Microbisporicin) and NVB302 (7-aminoheptylamido-deoxyactagardine B) in the discussion section in lines 385-388 and added one figure in the supplementary information (please see below), which are representative lanthipeptides that have advanced to preclinical development stages.

“In comparison with gallidermin, NAI-107, and NVB302, which have advanced to preclinical development stages, massatide A features relatively smaller molecular weight and more concise crosslinks (Figure S40). Its antibacterial activity against gram-positive bacteria was superior to that of gallidermin and NVB302.”

Compound	Structural features	Activity	Synthesis																	
 Massatide A	 Two thioethers AviCys motif D-amino acid N-di-methylation MW = 1635 Da 	   strains MIC (ug/ml)     G+ S. aureus ATCC25923 0.5   B. subtilis 168 2   M. luteus DSM11790 0.05   G- Escherichia coli DH5a >128   Pseudomonas aeruginosa PAO1 >128   	strains		MIC (ug/ml)	G+	S. aureus ATCC25923	0.5	B. subtilis 168	2	M. luteus DSM11790	0.05	G-	Escherichia coli DH5a	>128	Pseudomonas aeruginosa PAO1	>128	biosynthesis		
strains		MIC (ug/ml)																		
G+	S. aureus ATCC25923	0.5																		
	B. subtilis 168	2																		
	M. luteus DSM11790	0.05																		
G-	Escherichia coli DH5a	>128																		
	Pseudomonas aeruginosa PAO1	>128																		
 gallidermin	 Four thioethers AviCys motif D-amino acid Positively charged residues MW = 2166 Da 	   strains MIC (ug/ml)     G+ S. aureus SG511 4   S. aureus E88 8   M. luteus ATCC9341 0.25   M. luteus 15957 0.5   G- Escherichia coli ATCC11775 128   Pseudomonas aeruginosa BC19 128   	strains		MIC (ug/ml)	G+	S. aureus SG511	4	S. aureus E88	8	M. luteus ATCC9341	0.25	M. luteus 15957	0.5	G-	Escherichia coli ATCC11775	128	Pseudomonas aeruginosa BC19	128	biosynthesis
strains		MIC (ug/ml)																		
G+	S. aureus SG511	4																		
	S. aureus E88	8																		
	M. luteus ATCC9341	0.25																		
	M. luteus 15957	0.5																		
G-	Escherichia coli ATCC11775	128																		
	Pseudomonas aeruginosa BC19	128																		
 Microbisporicin/NAI-107 Pro* = 3,4-dihydroxy-proline, Microbisporicin A1 Pro* = 4-hydroxy-proline, Microbisporicin A2	 Five thioethers AviCys motif D-amino acid Halogenation MW = A1, 2246 Da A2, 2230 Da 	   strains MIC (ug/ml)     G+ S. aureus ATCC538P <0.13   S. aureus VISA 2   E. faecalis 1   E. faecalis (van-sensitive) 1   G- Escherichia coli >128   	strains		MIC (ug/ml)	G+	S. aureus ATCC538P	<0.13	S. aureus VISA	2	E. faecalis	1	E. faecalis (van-sensitive)	1	G-	Escherichia coli	>128	biosynthesis		
strains		MIC (ug/ml)																		
G+	S. aureus ATCC538P	<0.13																		
	S. aureus VISA	2																		
	E. faecalis	1																		
	E. faecalis (van-sensitive)	1																		
G-	Escherichia coli	>128																		
 NVB302 (7-aminoheptylamido-deoxyactagardine B)	 Four thioethers D-amino acid Halogenation 7-aminoheptylamido substitution MW = 1971 Da 	   strains MIC (ug/ml)     G+ S. aureus MSSA 8-32   S. aureus MRSA 8-32   E. faecalis (van-sensitive) 4-8   E. faecalis VREF 2-16   S. epidermidis 2-32        	strains		MIC (ug/ml)	G+	S. aureus MSSA	8-32	S. aureus MRSA	8-32	E. faecalis (van-sensitive)	4-8	E. faecalis VREF	2-16	S. epidermidis	2-32				Chemical Semi-synthesis
strains		MIC (ug/ml)																		
G+	S. aureus MSSA	8-32																		
	S. aureus MRSA	8-32																		
	E. faecalis (van-sensitive)	4-8																		
	E. faecalis VREF	2-16																		
	S. epidermidis	2-32																		

“Also, there is no discussion about the possible mechanism of antibacterial activity.”

We have included a discussion on the proposed mode of action of massatide A in the discussion section in lines 388-394 of our manuscript. “**Avi(Me)Cys-containing lanthipeptides are reported to target lipid II, an essential precursor in cell wall biosynthesis. One characterized ACyP is cacaoidin, which displays a dual mechanism of action by binding to lipid II and interacting with cell wall transglycosylase. Massatide A, which shares structural similarities with cacaoidin, may potentially target lipid II or certain membrane proteins involved in peptidoglycan biosynthesis. However, a comprehensive study is necessary to fully uncover the detailed mode of action of massatide A, which will be the primary focus of our ongoing research.**”

In our ongoing study, we have obtained some preliminary unpublished data supporting our hypothesis that massatide A may inhibit cell wall biosynthesis. We assessed UDP-N-acetyl-muramic acid pentapeptide levels, an essential precursor in cell wall biosynthesis, in the *S. aureus* ATCC259923 strain treated with or without massatide A. Comparative HPLC-HRMS analysis revealed its increased levels in the massatide-treated strain (Figure A-C below, unpublished data), suggesting potential impairment of cell wall biosynthesis due to hindered precursor utilization. The mode of action of massatide A will be further investigated in our ongoing research.

11. p.15, last line: the term "efficacy" is used for animal studies, not for in vitro studies

Response 11: We revised this sentence to “**Massatide A demonstrated a broad-spectrum antibacterial activity against gram-positive bacteria**” in lines 295-296.

12. p. 16: the data that sistertide A1 and A2 act synergistically are not convincing - see also main comment above.

Response 12: Thanks for your suggestion. We acknowledge the concern regarding the insufficient evidence supporting the presence of a synergistic effect in our study. As a result, we have opted to remove any conclusions relating to synergy.

13. p.17, middle paragraph: "immense potential" is not appropriate

Response 13: Thank you for your suggestion. We have revised the sentence in lines 335-336 to “**showcasing its potential as a therapeutic agent against gram-positive pathogens**”.

14. Discussion, first paragraph: "We successfully discovered five ACPs that exhibited potent antibacterial activity". Actually, only massatides do.

Response 14: We revised this sentence in line 362 to "We successfully discovered one ACyP, massatide A, that exhibited potent antibacterial activity against a wide range of gram-positive pathogens".

15. The sentence "Among the uncharacterized BGCs in our dataset, we identified certain BGCs containing a protein with a DUF3105 domain (Figure S39).", if important, should be part of Results and properly explained, not thrown in in the Discussion.

Response 15: We appreciate your suggestions and have decided to remove the statement that is not essential to the manuscript to enhance the coherence of the discussion. The revised sentences in lines 363-367 are as follows: "However, it is worth noting that our approach may have overlooked potential novel compounds with unanticipated modifications that were not predictable by rule-based metabolomic analysis. The presence of unknown modification enzymes associated with BGCs suggests the need for enhanced strategies in enzyme function annotation and metabolomics analysis in rule-based omics mining."

16. In the Discussion, the authors refer to water solubility issues for massatide A, but have not shown any data.

Response 16: Thank you for bringing this to our attention. We were not able to quantitatively measure the solubility of massatide A. Our observation of adding 300 μ l of water to 0.9 mg of massatide A indicated that the compound was not fully dissolved and left some precipitate at the bottom of the tube. We understand that this observation alone may not be sufficient to conclusively determine the water solubility of massatide A. Therefore, to avoid any confusion, we have decided to remove the mention of "water solubility" in lines 398-399 from our manuscript: "there is an ongoing need for further structural enhancements to refine its drug-like characteristics, including safety profiles, especially regarding its potential toxicity in human kidney cells."

17. What was the concentration of massatide A before administration to animals?

Response 17: The concentration of massatide A stock solution (in 5% DMSO, 5% Tween-80, 90% saline) is 2.5 mg/ml (for 50 mg/kg group), 1.5 mg/ml (for 30 mg/kg group) and 0.5 mg/ml (for 10mg/kg group). We injected 0.4 ml solution to each mouse. We added this sentence in lines 482-484, method part (mouse septicaemia model) of the main text.

18. How did the authors establish chirality from the data reported in Tables S8–S11.

Response 18: We conducted Advance Marfey's analysis of compounds **1**, **2**, **4** and **5** (Tables S8-S11) to establish the D- and L-configurations of each amino acid, following the well-established Marfey method (ref: doi.org/10.1021/ac970289b). Briefly, after hydrolysing each peptide using 6M HCl, the resulted mixture will separately react with D-FDLA and L-FDLA. For an amino acid X, the derivatives will be X-L-FDLA and X-D-FDLA. We can infer X based on:

- (1) the m/z determine the amino acid type and
- (2) the retention time, if X-L-FDLA < X-D-FDLA, then X is L-configured. Otherwise, X is D-configured. Exceptions are argine, lysin and histidine, which exhibit opposite retention time as shown below.

Amino acid	Elution order	Configuration
Ala	Ala-L-FDLA → Ala-D-FDLA	L
	Ala-D-FDLA → Ala-L-FDLA	D
Val	Val-L-FDLA → Val-D-FDLA	L
	Val-D-FDLA → Val-L-FDLA	D
Leu	Leu-L-FDLA → Leu-D-FDLA	L
	Leu-D-FDLA → Leu-L-FDLA	D
Ile	Ile-L-FDLA → Ile-D-FDLA	L
	Ile-D-FDLA → Ile-L-FDLA	D
Met	Met-L-FDLA → Met-D-FDLA	L
	Met-D-FDLA → Met-L-FDLA	D
Phe	Phe-L-FDLA → Phe-D-FDLA	L
	Phe-D-FDLA → Phe-L-FDLA	D
Tyr	Tyr-L-FDLA(mono-α) → Tyr-D-FDLA(mono-α)	L
	Tyr-L-FDLA(di) → Tyr-D-FDLA(di)	
	Tyr-D-FDLA(mono-α) → Tyr-L-FDLA(mono-α)	D
	Tyr-D-FDLA(di) → Tyr-L-FDLA(di)	
Pro	Pro-L-FDLA → Pro-D-FDLA	L
	Pro-D-FDLA → Pro-L-FDLA	D
Ser	Ser-L-FDLA → Ser-D-FDLA	L
	Ser-D-FDLA → Ser-L-FDLA	D
Thr	Thr-L-FDLA → Thr-D-FDLA	L
	Thr-D-FDLA → Thr-L-FDLA	D
Glu	Glu-L-FDLA → Glu-D-FDLA	L
	Glu-D-FDLA → Glu-L-FDLA	D
Asp	Asp-L-FDLA → Asp-D-FDLA	L
	Asp-D-FDLA → Asp-L-FDLA	D
Gln	Gln-L-FDLA → Gln-D-FDLA	L
	Gln-D-FDLA → Gln-L-FDLA	D
Asn	Asn-L-FDLA → Asn-D-FDLA	The diastereomers were not resolved
	Asn-D-FDLA → Asn-L-FDLA	
Lys	Lys-L-FDLA(mono-α) → Lys-D-FDLA(mono-α)	D
	Lys-D-FDLA(di) → Lys-L-FDLA(di)	
	Lys-D-FDLA(mono-α) → Lys-L-FDLA(mono-α)	L
	Lys-L-FDLA(di) → Lys-D-FDLA(di)	
His	His-L-FDLA(mono-α) → His-D-FDLA(mono-α)	D
	His-D-FDLA(di) → His-L-FDLA(di)	
	His-D-FDLA(mono-α) → His-L-FDLA(mono-α)	L
	His-L-FDLA(di) → His-D-FDLA(di)	
Arg	Arg-L-FDLA → Arg-D-FDLA	D
	Arg-D-FDLA → Arg-L-FDLA	L

Reviewer #3

The authors describe the heterologous expression and purification of five macrocyclic peptides. Thereof, two compounds, the massatides A and B, showed promising MIC values against Gram-positive bacteria. Noteworthy, massatide A was tested in vivo using a mouse septicemia model, in which the infection was caused by MRSA. The authors screened genome data to select putative biosynthetic gene clusters (BGCs) related to known Aminovinyl-(methyl)-cysteine-containing cyclopeptides, which are RiPPs. The in silico screen of available whole genome data showed that only a low percentage of precursor molecules is to date experimentally described. They selected three different BGCs and analyzed the wild type strains for production of the predicted metabolites. In two out of the three cases, the metabolites could be directly dereplicated from the crude extracts. Furthermore, they successfully heterologously expressed the three BGCs in a *Streptomyces albus* host. This enabled purification of metabolites corresponding to each of the three BGCs.

Overall, the manuscript describes nicely a natural product discovery workflow for the detection of potential lead structures for future antibiotic development. By reading the manuscript, this reviewer had the feeling that a few points could be considered to make the statements more clear, or to adjust the wording a bit.

Response: We appreciate both positive comments and informative suggestions from this reviewer. Please find below our point-by-point responses, which we believe address all the comments raised.

1. In the introduction section, it is stated that the scientific community has turned its attention to antimicrobial peptides (AMPs) as a potential solution for new antibiotics in addition to small molecules. The definition of AMPs is not clear and causes difficulties. My feeling would be that usually AMPs are regarded as ribosomally synthesized peptides with a certain length (e.g., depending on the definition colistin might be a member or not....mostly people would even not directly think about thiostrepton). Therefore, the smaller molecules, e.g. darobactin, dynobactin, as well as the nonribosomal peptides mentioned are not fitting. It could be better to directly refer to the term “macrocyclic peptides” used in the next sentence.

Response 1: Antimicrobial peptides (AMP) represent a large family of peptides that exhibit antimicrobial activities (<https://doi.org/10.1016/j.bcp.2016.09.018>). AMP usually refers to ribosomal peptide consist of between 10 and around 50 amino acid residues (<https://doi.org/10.1016/j.cub.2015.11.017>), with most of them being linear peptides. The RiPPs we are studying are macrocyclic peptides featuring thioether and AviCys moieties. Therefore, we have revised “antimicrobial peptides” to “**macrocyclic peptides**” in the introduction section. We agree that it is more appropriate to use “macrocyclic peptides” in our manuscript.

2. Then, it is stated that the grind and find approach is inadequate. This is a strong word,

since the antibiotics in use were discovered that way, and the work presented by the authors could be also seen as the same approach. Some compounds were produced and then tested for their bioactivity. Of course, I totally agree that a rationale-based prioritization is needed to enhance probability of success.

Response 2: Thank you for the suggestion. We have removed “grind and find approach” and revised the original statement to “The traditional chemical and/or biological screening method in uncovering new antibiotics from field-collected resources or laboratory-cultured organisms has proven to be both time-consuming and inadequate” in lines 50-51.

3. The abbreviation “ACPs” could be reconsidered (ACyPs?), since with the NRPs also modular systems are mentioned and there ACPs could be acyl carrier proteins.

Response 3: Thanks for your suggestion. We have revised the abbreviation “ACPs” to “ACyPs”.

4. The advantage of the mass mapping pipeline is not directly clear. It is stated that this helped to swiftly identify the relevant BGC before isolation. However, the authors used heterologous expression to isolate the metabolites of interest. Then, the BGC is already clear. Out of curiosity, were there many metabolites with such a high molecular weight present in the extracts and so many putative BGCs present in the genomes that it is unclear if the metabolite can be linked to a corresponding BGC by MS/MS analysis?

Response 4: The advantage of the mass mapping pipeline is its ability to swiftly link mass signals to potential BGCs in the wild type strains. Additionally, it enables the calculation of modification types for each amino acid based on tandem mass data. This information can directly inform the putative structure of the final RiPP compound. In contrast, heterologous expression of BGCs poses significantly more challenges and is time-consuming, with outcomes sometimes yielding no production. Therefore, our initial approach involves utilizing the mass mapping pipeline to identify the target product in wild-type strains. Subsequently, heterologous expression serves to further validate and confirm the results obtained from the mass mapping pipeline. This integrated strategy enhances both the efficiency and the reliability of identifying and characterizing RiPP compounds.

We indeed observed some metabolites with high molecular weight in the extracts. For example, in the extracts of wild type strain DSM42122 that harbor many BGCs, there are 7 obvious peaks that are larger than 1000 Da (**SI, Supplementary note, Mass calculation and mapping workflow**). We then mapped all these putative peaks with calculated mass data to find potential targets. If both observed MS1 and MS2 data align well with the calculated data, we can confidently link the BGC with the corresponding metabolites.

5. Furthermore, for the *keb* cluster the analysis of the wild type strain did not show any predicted metabolite. Then, it was directly used for heterologous expression. Reasonable to use heterologous approach, since it was working for the others. However, it somehow could break the logic of the workflow described in the manuscript.

Response 5: In this specific case, our initial analysis of the wild-type strain using the workflow revealed no discernible target peaks from the fermentation, suggesting that the *keb* cluster may be inactive in its native host. Consequently, we transitioned to a heterologous expression system to activate this BGC. Leveraging mass calculation, we efficiently located the target compound and proposed its structure in the heterologous host.

To enhance clarity, we have revised the sentence in lines 253-255 as follows: “Our initial analysis of the wild-type DSM42048 using the workflow revealed no discernible target peaks from the fermentation. Therefore, we hypothesized that the BGC might be either inactive or expressed at a very low level in the native host. Subsequently, we tried to heterologously express.....”

6. In the results section: The sentence “Given the presence of these two precursor subfamilies within one BGC” is somehow unclear. Are there always 2 precursors, one with 2 and one with 3 Cys residues? I agree with the point that “that a single maturation system within the *sis* family could generate ACPs with diverse thioether rings”; however, this is based on the specific precursors within one BGC (maybe the word families makes it hard to read as the sentence is structured right now).

Response 6: Yes, all the BGCs in the *sis* family contain at least 2 precursors, with one having 2 cysteines and another having 3 cysteines. We have removed the original statement “Given the presence of these two precursor subfamilies within one BGC” as it was unnecessary and potentially misleading. We believe that the revised sentence is easier to understand.

7. Is there any specific reason why only MatR1 and not MatR2 was analyzed experimentally?

Response 7: Previously, the construction of knockout plasmid pCZMAT_ΔR2 was unsuccessful. While this manuscript was under revision, we successfully constructed *S. albus*/pCZMAT_ΔR2 after several attempts. We hypothesize that this knockout strain would produce a mass tide A analog with D-Ala9 and D-Ala16 without reduction, which would have a calculated monoisotopic mass of 1629.74. However, upon analyzing the fermentation results, we did not observe the expected mass signal of this analog.

8. In the figures the crude extracts of the producer strains are not shown, only of the

heterologous producer strains. Is this due to the fact that the expression level is too low in the wild type strains? Maybe it could be mentioned how concentrated the tested extracts were (since the inhibition zones are quite small).

Response 8: Thank you for bringing this to our attention. We utilized the crude extract from the heterologous producer due to the presence of numerous other antibacterial compounds in the crude extract of the wild-type strain, which could potentially confound our observations. In contrast, the heterologous expression strain offers a much cleaner background. Additionally, we were able to establish a control group (host with empty vector) to compare with the heterologous strain, thereby highlighting the antibacterial activity originating from the targeted BGC.

Nevertheless, we have included the High-Resolution Mass Spectrometry (HRMS) profile of the wild-type extract in the Supporting Information Figure S3 and S14, demonstrating that the wild-type strain can still produce the same metabolites generated by the BGC.

In Figure 3e (i) and (ii), we did not concentrate the sample and directly utilized fermentation agar with corresponding strains to test the activity. In Figure 4a, the crude extract was concentrated tenfold from fermentation agar.

9. “one tailoring enzyme system constructs multiple crosslinking patterns”. Can this be stated in this way? Is the enzyme identified in the different BGCs identical, or homologous?

Response 9: Our statement is appropriate as supported by the following evidence: 1) BGCs from the *sis* family contains at least two precursors, one with two cysteines and one with three cysteines. 2) In the characterized candidates in this manuscript, the maturases from the BGCs respectively modify the two-cysteine and three-cysteine precursors to form two-thioether and three-thioether compounds. The maturases identified in the different BGCs are homologous. Below are a few representative BGCs from the *sis* family.

10. Antibacterial activity. It is stated that “slightly larger inhibition zone in Figure 4e, suggesting a synergistic effect”. Could be somehow contradictory. Is it clear that it is a synergistic and not an additive effect? (The same for kebanetides A1 and A2)

This point should be clarified throughout the manuscript. A checkerboard assay using the purified compounds could be done.

Response 10: Thanks for the suggestion. We acknowledge the recommendation to conduct checkerboard assays using compounds from the *sis* BGC (**2** and **3**) or the *keb* BGC (**4** and **5**) to explore potential synergistic effects. Unfortunately, the quantities of compounds **3-5** available to us are inadequate for conducting the checkerboard assay. Despite isolating 4 mg of compound **2** from an 8 L fermentation, we encountered challenges in obtaining an adequate amount of compound **3** due to its low yield. Similarly, our efforts yielded only 0.2 mg of compound **4** from a 2 L fermentation, which barely met the requirements for the MIC assay. As a result of the limited material available for the checkerboard assay, we have decided to remove any conclusions regarding the potential synergistic relationship between **2** and **3** or **4** and **5**.

11. “potential as a promising candidate for gram-positive antibiotics.” Might be a far fetch at this stage of the project. Maybe it can be toned down a bit.

Response 11: Yes, we acknowledge that the statement may be overstated at this stage. We thus delete “promising” in line 298.

12. Efficacy in vitro. I understand the point that only an 8-fold increase on the MIC was reached within 6 days. However, absolutely no shift was seen for the control vancomycin, even so resistance is reported and should occur in *S. aureus*. (Some numbers in an old article from 2002 “An Elevated Mutation Frequency Favors Development of Vancomycin Resistance in *Staphylococcus aureus*”). Maybe some more words as explanation could help here.

Anything known about the mode of action? (I assume it will be related to other members of the Aminovinyl-(methyl)-cysteine-containing cyclopeptides)

Response 12: We highly value the reviewer's perceptive interpretation of the data. The serial passaging method we employed highlights the potential risk of spontaneous point mutations in genes (e.g., target proteins, regulators, and transporters). It is worth noting that vancomycin does not exhibit a shift in activity due to its bactericidal mechanism, which is based on binding to the bacterial cell envelope rather than a protein target like most antibiotics. Vancomycin interacts with the D-Ala-D-Ala region of Lipid II. However, resistance to vancomycin can arise from the substitution of D-Ala-D-lac or D-Ala-D-Ser in lipid II by enzymes VanH, A, and X, which reduces vancomycin's affinity (source: <https://onlinelibrary.wiley.com/doi/epdf/10.1002/pro.3819>). Regarding massatide A,

the absence of significant changes in the assay suggests a relatively low risk of resistance caused by point mutations. However, it is important to note that resistance may still occur if the structure of its target in bacteria undergoes modifications. To clarify, we have included an additional sentence in lines 394-397 in the discussion section. "It is worth noting that vancomycin-resistant strains have been reported since 1986, despite the absence of a shift in our experiment (Figure 5d). Therefore, it is crucial to take into account the potential development of resistance to massatide A with prolonged usage, especially if modifications occur to the structure of its putative target, such as lipid II, in bacteria."

The known mode of action of several known ACyPs is that they will bind to lipid II and inhibit cell wall biosynthesis (<https://pubs.acs.org/doi/epdf/10.1021/jacsau.1c00308>). We have included a discussion on the proposed mode of action of massatide A in lines 388-394 in the discussion section of our manuscript. "Avi(Me)Cys-containing lanthipeptides are reported to target lipid II, an essential precursor in cell wall biosynthesis. One characterized ACyP is cacaoidin, which displays a dual mechanism of action by binding to lipid II and interacting with cell wall transglycosylase. Massatide A, which shares structural similarities with cacaoidin, may potentially target lipid II or certain membrane proteins involved in peptidoglycan biosynthesis. However, a comprehensive study is necessary to fully uncover the detailed mode of action of massatide A, which will be the primary focus of our ongoing research."

13. "massatide A showed low toxicity to Hela cells only at a high concentration of 200 µg/mL". Is this wording correct? From 40 to 200 µg/mL the viability of Hela cells dropped from 1.0%? to around 0.25 (Figure S38). Can this be regarded as low? Furthermore, the cytotox against Hek293T cells looks stronger, even at 8 µg/mL a clear decrease in viability. This would indicate a rather small therapeutic window. In Figure S38 E it seems like the values were exchanged between massatide A and B. Furthermore, is the IC50 against Hela cells indeed >200 (see point before)?

Response 13: We appreciate the reviewer for bringing to our attention the potential misleading statement. To provide a more precise and accurate description, we have revised the statement in lines 323-324 as follows: "Massatide A exhibits cytotoxicity against Hela cells, with an IC50 value of 136.3 µg/mL, while massatide B did not exhibit any significant cytotoxicity against Hela cells."

Regarding the previous symbol "/" in Figure E, we would like to clarify that it indicated that we were unable to calculate the IC50 value based on the previous data. To accurately determine the IC50 value of massatide A against Hela cells, we have modified the testing range to 1-200 µg/mL accordingly. The IC50 of massatide A against Hela was found to be 136.3 µg/mL, while the IC50 of massatide B against Hela was higher than 200 µg/mL. We have updated the Supplementary Figure S39 to reflect these changes, and it is provided below for reference.

Before:

revised:

It is important to note that, like many peptide antibiotics, massatide A exhibits relatively higher toxicity towards Hek293T cells compared to Hela cells. This limitation should be considered when assessing the potential of massatide A. Addressing this limitation may require structural optimization and bioengineering of massatide in future endeavors.

14. Figure 5. Here it would be nice to get some later time points. In a) it can be seen that the curve for the 1xMIC starts to rise as well. Even more important, why are the data missing in c)?

It would be highly interesting to see how the CFUs develop. Will there be a rebound? Will there be resistance development visible in these experiments?

Response 14: We have incorporated additional data at later time points and made revisions to Figure 5. In Figure 5a, we have determined that it takes approximately 36 hours to reach the stationary phase. In the revised figure, we still observe a slight increase in bacterial growth after 24 hours. This suggests that the bacteria were not completely eradicated under this concentration. However, the growth rate is significantly inhibited by the compound, resulting in extremely slow growth.

Original Figure 5a:

Revised Figure 5a:

In figure 5c, we added 26 h time point.

Original Figure 5c:

Revised Figure 5c:

“It would be highly interesting to see how the CFUs develop. Will there be a rebound? Will there be resistance development visible in these experiments?” We did not observe the rebound in the time-killing assay within 26h and the living cells were continuously decreasing. Additionally, we observed that the culture medium remained clear even after 48 hours, leading us to conclude that there would be no rebound. Although we did not observe resistance in this experiment, it is worth noting that the time-killing assay usually tests for only 24 hours, which may not be enough time for resistance development to occur. In Figure 5d, we evaluated the risk of resistance and found that the activity remained stable for at least the first four days. This suggests that the risk of resistance developing in the short term is low.

15. In the discussion, the point “with low toxicity and minimal risk of resistance” is repeated. However, it is questionable if low tox is really the appropriate term.

Response 15: We initially mentioned that our compound exhibited low acute toxicity in mice. However, we have since discovered that compound 1 exhibits moderate kidney toxicity in Hek293T cells, making this statement inappropriate. Therefore, we have decided to delete this sentence.

16. In summary, the in vivo data are convincing. However, it would be good to know which compound levels were reached within the mouse. Maybe this could be used for the argumentation about the cytotox issues observed before (keeping the observed cytotox against liver cells in mind).

Response 16: In order to investigate compound levels in vivo (pharmacokinetics), blood samples must be collected at various time points. Due to the amount of sample required, rats weighing approximately 200g would need to be used instead of mice weighing around 20g. Additionally, three groups at doses of 10 mg/kg, 30 mg/kg, and 50 mg/kg would be necessary. However, even with the most simplified test, it would necessitate the tested compound to be on a hundred-milligram scale, which surpasses the currently available quantity. Therefore, chemical synthesis or metabolic engineering strategies would be required to obtain the necessary amount of the compound for future pharmacokinetics studies, which are beyond the scope of this manuscript.

Reviewer #4

Response: We appreciate both positive comments and informative suggestions from this reviewer.

REVIEWERS' COMMENTS

Reviewer #2 (Remarks to the Author):

The authors have done a thorough job in addressing my comments (and, in my opinion, those by the other reviewers). The result manuscript is greatly improved and there are no further suggestions from my side.

Reviewer #3 (Remarks to the Author):

In the revised version, the authors tried to address the points raised by the reviewers. The overstated phrases had been toned down and the clarity of the manuscript was improved.

Overall, the workflow presented is sound and a new bioactive compound was discovered and tested in vitro and in vivo.

Reviewer #4 (Remarks to the Author):

“I co-reviewed this manuscript with one of the reviewers who provided the listed reports. This is part of the Nature Communications initiative to facilitate training in peer review and to provide appropriate recognition for Early Career Researchers who co-review manuscripts.”